# Repression by the *Arabidopsis* TOPLESS corepressor requires association with the core mediator complex

Alexander R Leydon[1], Wei Wang[2†], Hardik P Gala[1], Sabrina Gilmour[1], Samuel Juarez-Solis[1], Mollye L Zahler[1], Joseph E Zemke[1], Ning Zheng[2,3], Jennifer L Nemhauser[1]*

[1]Department of Biology, University of Washington, Seattle, United States; [2]Department of Pharmacology, Seattle, United States; [3]Howard Hughes Medical Institute, University of Washington, Seattle, United States

**Abstract** The plant corepressor TOPLESS (TPL) is recruited to a large number of loci that are selectively induced in response to developmental or environmental cues, yet the mechanisms by which it inhibits expression in the absence of these stimuli are poorly understood. Previously, we had used the N-terminus of *Arabidopsis thaliana* TPL to enable repression of a synthetic auxin response circuit in *Saccharomyces cerevisiae* (yeast). Here, we leveraged the yeast system to interrogate the relationship between TPL structure and function, specifically scanning for repression domains. We identified a potent repression domain in Helix 8 located within the CRA domain, which directly interacted with the Mediator middle module subunits Med21 and Med10. Interactions between TPL and Mediator were required to fully repress transcription in both yeast and plants. In contrast, we found that multimer formation, a conserved feature of many corepressors, had minimal influence on the repression strength of TPL.

**\*For correspondence:**
jn7@uw.edu

**Present address:** [†]Key Laboratory of Plant Stress Biology, State Key Laboratory of Cotton Biology, School of Life Science, Jinming Campus, Henan University, Kaifeng, China

**Competing interests:** The authors declare that no competing interests exist.

## Introduction

Control over gene expression is essential for life. This is especially evident during development when the switching of genes between active and repressed states drives fate determination. Mutations that interfere with repression lead to or exacerbate numerous cancers (*Wong et al., 2014*) and cause developmental defects in diverse organisms (*Grbavec et al., 1998*; *Long et al., 2006*), yet many questions remain about how cells induce, maintain, and relieve transcriptional repression. Transcriptional repression is controlled in part by a class of proteins known as corepressors that interact with DNA-binding transcription factors and actively recruit repressive machinery. Transcriptional corepressors are found in all eukaryotes and include the animal SMRT (silencing mediator of retinoic acid and thyroid hormone receptor) and NCoR (nuclear receptor corepressor) complexes (*Mottis et al., 2013*; *Oberoi et al., 2011*), the yeast Tup1 (*Keleher et al., 1992*; *Matsumura et al., 2012*; *Tzamarias and Struhl, 1994*), and its homologs *Drosophila* Groucho (Gro) and mammalian transducing-like enhancer (TLE) (*Agarwal et al., 2015*).

In plants, the role of Gro/TLE-type corepressors is played by TOPLESS (TPL), TOPLESS-RELATED (TPR1-4), LEUNIG (LUG) and its homolog (LUH), and High Expression of Osmotically responsive genes 15 (HOS15) (*Causier et al., 2012*; *Lee and Golz, 2012*; *Liu and Karmarkar, 2008*; *Long et al., 2006*; *Zhu et al., 2008*). Plant corepressors share a general structure, where at the N-terminus a LIS1 homology (LisH) domain contributes to protein dimerization (*Delto et al., 2015*; *Kim et al., 2004*). At the C-terminus, WD40 repeats form beta-propeller structures that are involved in protein-protein interactions (*Collins et al., 2019*; *Liu et al., 2019*). In TPL family corepressors, the LisH is followed by a C-terminal to LisH (CTLH) domain that binds transcriptional repressors through

an Ethylene-responsive element binding factor-associated Amphiphilic Repression (EAR) motif found in partner proteins (*Causier et al., 2012*; *Kagale et al., 2010*). The N-terminal domain also contains a CT11-RanBPM (CRA) domain, which provides a second TPL dimerization interface and stabilizes the LisH domain (*Ke et al., 2015*; *Martin-Arevalillo et al., 2017*). The beta-propellers bind to the non-EAR TPL recruitment motifs found in a subset of transcriptional regulators (RLFGV- and DLN-type motifs; *Collins et al., 2019*; *Liu et al., 2019*) and may control protein interaction with other repressive machinery. Defects in the TPL family have been linked to aberrant stem cell homeostasis (*Busch et al., 2010*), organ development (*Gonzalez et al., 2015*), and hormone signaling (*Causier et al., 2012*; *Kagale et al., 2010*), especially the plant hormone auxin (*Long et al., 2006*).

We have previously demonstrated the recapitulation of the auxin response pathway in *Saccharomyces cerevisiae* (yeast) by porting individual components of the *Arabidopsis* auxin nuclear response (*Pierre-Jerome et al., 2014*). In this <u>*Arabidopsis thaliana*</u> <u>A</u>uxin <u>R</u>esponse <u>C</u>ircuit in *Saccharomyces cerevisiae* (*At*ARC^Sc; *Figure 1A*), an auxin-responsive transcription factor (ARF) binds to a promoter driving a fluorescent reporter. In the absence of auxin, the ARF protein activity is repressed by interaction with a full-length Aux/IAA protein fused to the N-terminal domain of TPL. Upon the addition of auxin, the TPL-IAA fusion protein is targeted for degradation through interaction with a member of the Auxin Signaling F-box protein family and releases the transcriptional repression of the fluorescent reporter. Reporter activation can be quantified after auxin addition by microscopy or flow cytometry (*Pierre-Jerome et al., 2014*). In the original build and characterization of *At*ARC^Sc, it was noted that the two N-terminal truncations of TPL (N100 or N300) behave differently (*Pierre-Jerome et al., 2014*). While both truncations are able to repress the function of a transcriptional activator fused to an Aux/IAA, only the TPLN100 fusion shows alleviation of repression after auxin addition. TPLN300 fusions to Aux/IAAs maintain strong durable repression even under high concentrations of auxin. This disparity is not due to differential rates of protein degradation as both proteins appear to be turned over with equal efficiency after auxin addition (*Pierre-Jerome et al., 2014*).

The conservation of TPL's repressive function in yeast suggests that the protein partners that enact repression are conserved across eukaryotes. Consistent with this speculation, the series of alpha-helices that form the N-terminal portion of TPL (*Figure 1B*; *Martin-Arevalillo et al., 2017*) is highly reminiscent of naturally occurring truncated forms of mammalian TLE (*Gasperowicz and Otto, 2005*), such as Amino-terminal Enhancer of Split (AES) (*Zhang et al., 2010*), the Groucho ortholog LSY-22 (*Flowers et al., 2010*), and the unrelated mouse repressor protein MXI1 (*Schreiber-Agus et al., 1995*). Gro/TLE family members are generally considered to repress by recruiting histone deacetylases (HDACs) to control chromatin compaction and availability for transcription (*Chen and Courey, 2000*; *Long et al., 2006*). An alternative hypothesis has been described for Tup1 in yeast, where Tup1 blocks the recruitment of RNA polymerase II (Pol II) (*Wong and Struhl, 2011*), possibly through contacts with Mediator complex subunits Med21 or Med3 (*Gromöller and Lehming, 2000*; *Papamichos-Chronakis et al., 2000*). However, like many of these family members, multiple repression mechanisms have been described for TPL at different genetic loci. For example, TPL has been found to recruit the repressive CDK8 Mediator complex (*Ito et al., 2016*), chromatin remodeling enzymes such as Histone Deacetylase 19 (HD19) (*Long et al., 2006*), and directly bind to histone proteins (*Ma et al., 2017*).

Here, we leveraged the power of yeast genetics to interrogate the mechanism of TPL repression. Using *At*ARC^Sc, we discovered that the N-terminal domain of TPL contains two distinct repression domains that can act independently. We mapped the first, weaker repression domain to the first 18 amino acids of the LisH domain (Helix 1), and the second, more potent domain to Helix 8, which falls within the CRA domain. Full repression by Helix 8 required the Mediator complex, specifically direct interaction with Med21 and Med10. The Med21 residues that interact with TPL are the same ones required for transcriptional repression by the yeast corepressor Tup1. In addition, we found that multimerization of TPL was not required for repression in yeast or in plants. Our yeast results were validated with plant assays and extended to include evidence that interaction with the middle domain of Mediator was required for TPL repression of the auxin-regulated development program giving rise to lateral roots. Our findings point to a conserved functional connection between Tup1/TPL corepressors and the Mediator complex that together create a repressed state poised for rapid activation.

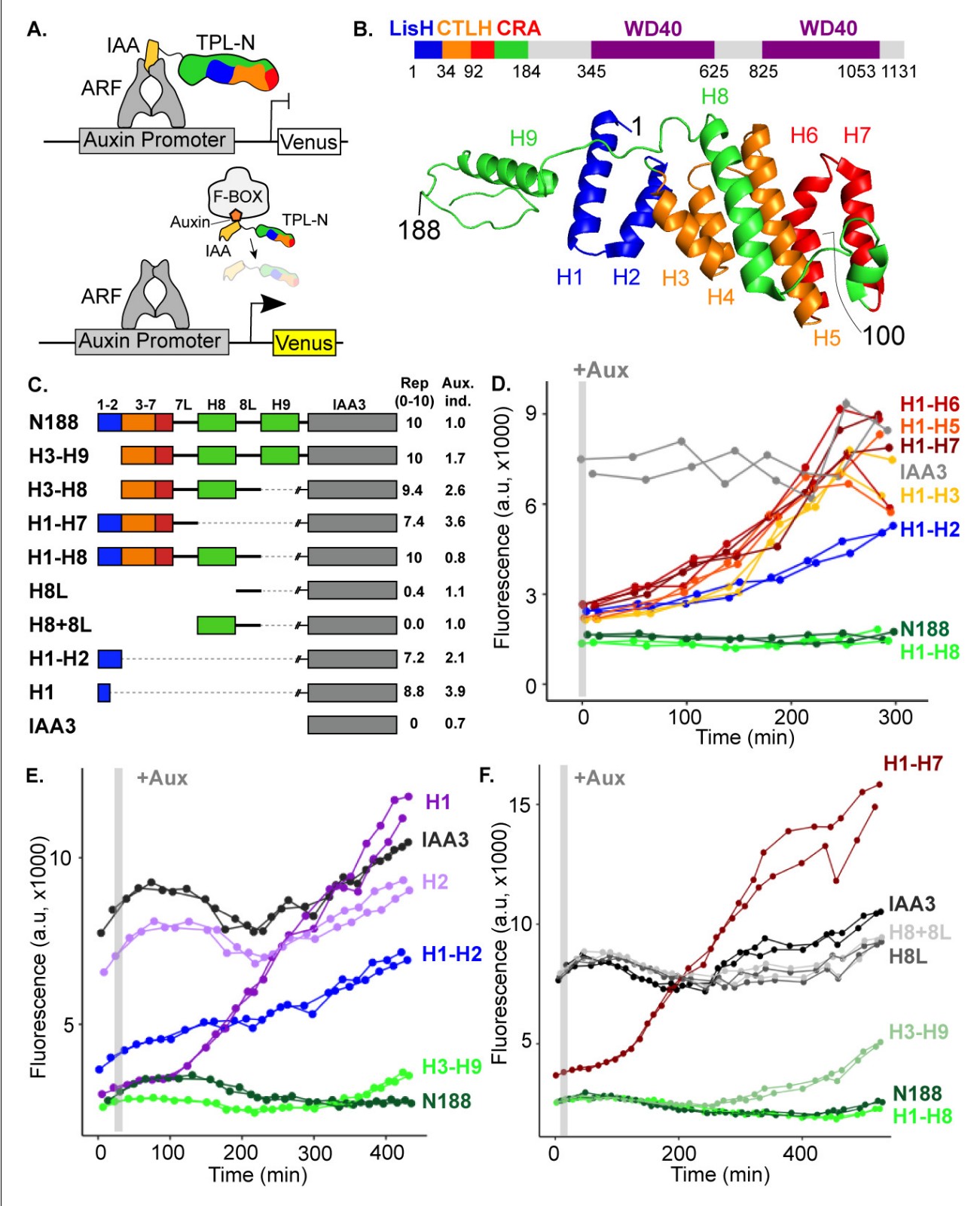

**Figure 1.** The N-terminal domain of TPL contains two independent repression domains. (A) Schematic of the ARC^Sc. The auxin-responsive promoter driving the fluorescent protein Venus carries binding sites for the auxin-responsive transcription factor (ARF). In the absence of auxin, the IAA-TPL-N fusion protein is bound to the ARF and maintains the circuit in a repressed state. Upon addition of auxin, the IAA-TPL protein is targeted for ubiquitination and subsequent protein degradation, activating transcription of the fluorescent reporter. (B) TPL domains are LisH (LIS1 homology motif,

*Figure 1 continued on next page*

*Figure 1 continued*

blue), CTLH (C-terminal LisH motif, orange), CRA (CT11-RanBPM; red, dimerization; green, foldback), and two WD40, beta-propeller motifs (purple). N-terminal domains are indicated on the solved structure of the first 202 amino acids (*Martin-Arevalillo et al., 2017*, 5NQS). The termini of the TPLN100 truncation used in the original ARC$^{Sc}$ studies is indicated. (**C**) Diagram indicating the structure of constructs analyzed in experiments shown in subsequent panels. For constructs with identical behavior (H1-H3, H1-H5, H1-H6, H1-H7), we included only a representative member (H1-H7) for simplicity. Repression Index (Rep.) is a scaled measure of repression strength with 0 set to the level of repression observed with IAA3 and 10 set to the level of repression by TPLN188. Auxin induction level (Aux. ind.) indicates the fold change difference between reporter expression before auxin addition (time zero) and at the end of an experiment (~500 min). (**D-F**) Helix 1 and the CRA domain (Helix 3–Helix 8) can act independently to repress transcription. Each panel represents two independent time-course flow cytometry experiments of the TPL helices indicated, all fused to IAA3. Every point represents the average fluorescence of 5–10,000 individually measured yeast cells (a.u.: arbitrary units). Auxin (IAA-10 µM) was added at the indicated time (gray bar, +Aux).

The online version of this article includes the following figure supplement(s) for figure 1:

**Figure supplement 1.** Helix 1 and the CRA domain (Helix 3-Helix 8) can act independently to repress transcription.

## Results

To understand how TPL represses transcription, we first localized repressive activity within the protein using the *At*ARC$^{Sc}$ (*Figure 1A*). The extent of auxin-induced turnover of TPLN100 and TPLN300 fusion proteins appears similar, although neither are completely degraded (*Pierre-Jerome et al., 2014*). In this way, auxin sensitizes the *At*ARC$^{Sc}$ to even subtle differences in the strength of repressive activity by reducing the relative concentration of the TPL fusion proteins. To pinpoint the region conferring the strong repression of TPLN300, we generated a deletion series of the N-terminus guided by the available structural information (*Figure 1B, C*; *Ke et al., 2015*; *Martin-Arevalillo et al., 2017*).

We started by identifying a shorter truncation, TPLN188, which behaved identically to TPLN300 (*Figure 1D*, *Pierre-Jerome et al., 2014*). Subsequently, we deleted each alpha helical domain starting with Helix 9 (constructs are named in the format Helix x – Helix y or Hx-Hy). We found that Helix 8 was required for the maximum level of repression activity and for the maintenance of repression after auxin addition (*Figure 1D*). All constructs lacking Helix 8 retained the ability to repress transcription, but this repression was lifted in the presence of auxin (*Figure 1D*) as had been observed for the original TPLN100 construct (*Pierre-Jerome et al., 2014*). In addition to the repressive activity of Helix 8, further deletions revealed that the 18 amino acids of Helix 1 were sufficient to confer repression on their own (H1, *Figure 1E*). To test whether Helix 8 activity depended on Helix 1, we analyzed a construct consisting of Helix 3 through Helix 9 (H3-H9, *Figure 1E*), which was able to repress transcription. Thus, Helix 1 (LisH) and Helix 8 (CRA) could contribute to TPL-mediated repression on their own (*Figure 1D*).

To identify the minimal domain required for Helix 8-based repression, we generated additional deletions (*Figure 1C, F*, *Figure 1—figure supplement 1*). Helix 8 and the following linker were not sufficient for repression (*Figure 1F*), and removal of Helix 9 or of the linker between Helix 8 and Helix 9 slightly increased sensitivity to auxin compared to TPLN188 (H1-H8Δ8L, *Figure 1—figure supplement 1*). A deletion that removed both the LisH and Helix 8 repression domains (H3-H7) was only able to weakly repress reporter expression (*Figure 1—figure supplement 1*). These results demonstrate that Helix 8, in combination with the linker between Helix 8 and Helix 9 (which folds over Helix 1), was required for maintaining repression following addition of auxin. Moreover, the repressive activity of Helix 8 and the linker were only functional in the context of the larger Helix 3-Helix 8 truncation that carries the CTLH domain and a portion of the CRA domain.

To determine which of the many known or predicted TPL-binding partners could mediate the repression activity of Helix8, we identified known interactors with either TPL or other Gro/TLE corepressors, and then introduced the *Arabidopsis* homologs of these genes into the cytoplasmic split-ubiquitin system (cytoSUS) (*Asseck and Grefen, 2018*). We chose cytoSUS over yeast two-hybrid because in cytoSUS the interaction between target proteins takes place in the cytoplasm, and we had observed that the TPL N-terminus could repress activation of yeast two hybrid prototrophy reporters (*Figure 2—figure supplement 1A*). Putative direct interactors include HDACs (AtHDAC9, AtHDAC6; *Long et al., 2006*), Histone proteins (Histone H3, Histone H4; *Ma et al., 2017*), and the Mediator components MED13 (AtMED13; *Ito et al., 2016*) and MED21, which has been demonstrated to interact with Tup1, the yeast homolog of TPL (*Gromöller and Lehming, 2000*). We did

not observe any interactions between TPLN188 and the HDACs HDA6 and HDA9; the histone protein AtHIS4; or the Mediator subunit AtMED13 (*Figure 2A*, *Figure 2—figure supplement 1B*). HDAC interaction with TPL has been previously hypothesized to occur through indirect interactions with partner proteins (*Krogan et al., 2012*); however, direct interactions with histones and MED13 have been detected (*Ito et al., 2016*; *Ma et al., 2017*). The absence of interaction between TPLN188 and these proteins may be due to differences between methods or interaction interfaces in the C-terminal WD40 repeats.

Strong interaction was detected between TPLN188 and AtMED21, a component of the Mediator middle domain (*Figure 2A*). MED21 is one of the most highly conserved Mediator subunits (*Bourbon, 2008*) and has a particularly highly conserved N-terminus (*Figure 2—figure supplement 2A, C–E*). In yeast, Tup1 interacts with the first 31 amino acids of ScMed21, with the first 7 amino acids being absolutely required for interaction and transcriptional repression (*Gromöller and Lehming, 2000*). We observed that the equivalent truncation of AtMED21 (AtMED21-N31) was sufficient for interaction with TPLN188 (*Figure 2A*). We next created truncations of the N-terminal domain of AtMED21 to closely match those that had been made in yeast (*Figure 2B*) where deletion of the first five amino acids of ScMed21 (ScΔ5Med21) severely reduce the ability of the Mediator complex to co-purify with Pol II and CDK8 kinase complex (*Sato et al., 2016*). Interaction between TPLN188 and AtMED21 similarly required the first five amino acids of AtMED21 (*Figure 2B*), and, as in yeast, this was not a result of destabilization of the AtMED21 protein (*Figure 2B*). In fact, N-terminal Med21 deletions increased protein levels with no interaction with TPLN188; this results in high confidence that the Med21 N-terminus is required for interaction. AtMED21 interaction was specific to the Helix8-based repression domain as it interacted with TPLH3-H9 (*Figure 2E*), and not TPLH1-H7 (*Figure 2—figure supplement 1C*). Further screening of middle domain Mediator components identified an additional interaction with AtMED10B, the predominantly expressed MED10 isoform in *Arabidopsis* (*Klepikova et al., 2016*; *Figure 2C*). Interactions between TPLN188 and both AtMED21 and AtMED10B were confirmed by immunoprecipitation (*Figure 2D*).

The MED21 N-terminus lies in the hinge region of the middle module and has residues that are exterior facing and could be docking points for protein-protein interactions (*Figure 2—figure supplement 2C–E*). A manual juxtaposition of the yeast Mediator structure with the *Arabidopsis* TPL N-terminal structure shows that Helix 8 and the linker following face away from the tetramer and are therefore optimally placed to interact with Mediator components (*Figure 2F*). To pinpoint which residues of Helix 8 coordinate repression through interaction with MED21, we identified solution-facing amino acids (*Martin-Arevalillo et al., 2017*). We reasoned that such residues were most likely not involved in stabilizing the hydrophobic interactions between intra-TPL helical domains and could therefore be available to interact with partner proteins. Eight amino acids in Helix 8 were mutated to alanine in the context of the H3-H8-IAA3 fusion protein (*Figure 3A*, light green residues) to enable assessment of repression activity in the absence (*Figure 3B*) or presence (*Figure 3C*) of auxin. No single amino acid was essential for repression. Two mutations (R140A and K148A) slightly increased baseline expression of the reporter (*Figure 3B, C*). With the exception of E152A, which behaved similarly to controls, all of the mutations altered the stability of repression after auxin addition, either by increasing (S138A, V145A, E146A, K149A) or decreasing (I142A) the final fluorescence level (*Figure 3C*). Mutating E146 and K149 also increased the speed with which the reporter responded to auxin (*Figure 3C*), suggesting that these two neighboring residues could be a critical point of contact with co-repressive machinery. S138A had a small increase in auxin sensitivity, while I142 reduced auxin sensitivity (*Figure 3E*).

We next tested whether the residues in Helix 8 that were required for repression (V145, E146, K148, K149; *Figure 3A–C*) were also required for interaction with AtMED21. Single-alanine mutations of these four amino acids in the context of TPLN188 significantly reduced interaction with AtMED21, while the quadruple mutation (here called Quad[AAAA]) completely abrogated AtMED21 binding (*Figure 3D*). These mutations had little effect on interaction with AtMED10B (*Figure 2—figure supplement 1D*). Introduction of Quad[AAAA] mutations into the Helix 3 through Helix 8 context (H3-H8-Quad[AAAA]) in the *At*ARC[Sc] largely phenocopied a deletion of Helix 8 (yellow and pink, *Figure 3E*, compare to H3-H7, *Figure 1—figure supplement 1*). In contrast, TPLN188-Quad[AAAA] largely retained the repressive activity of wild-type N188 (red and black, *Figure 3E*), consistent with the observation that Helix 1 is sufficient for full repression (*Figure 1E*). These results indicate that

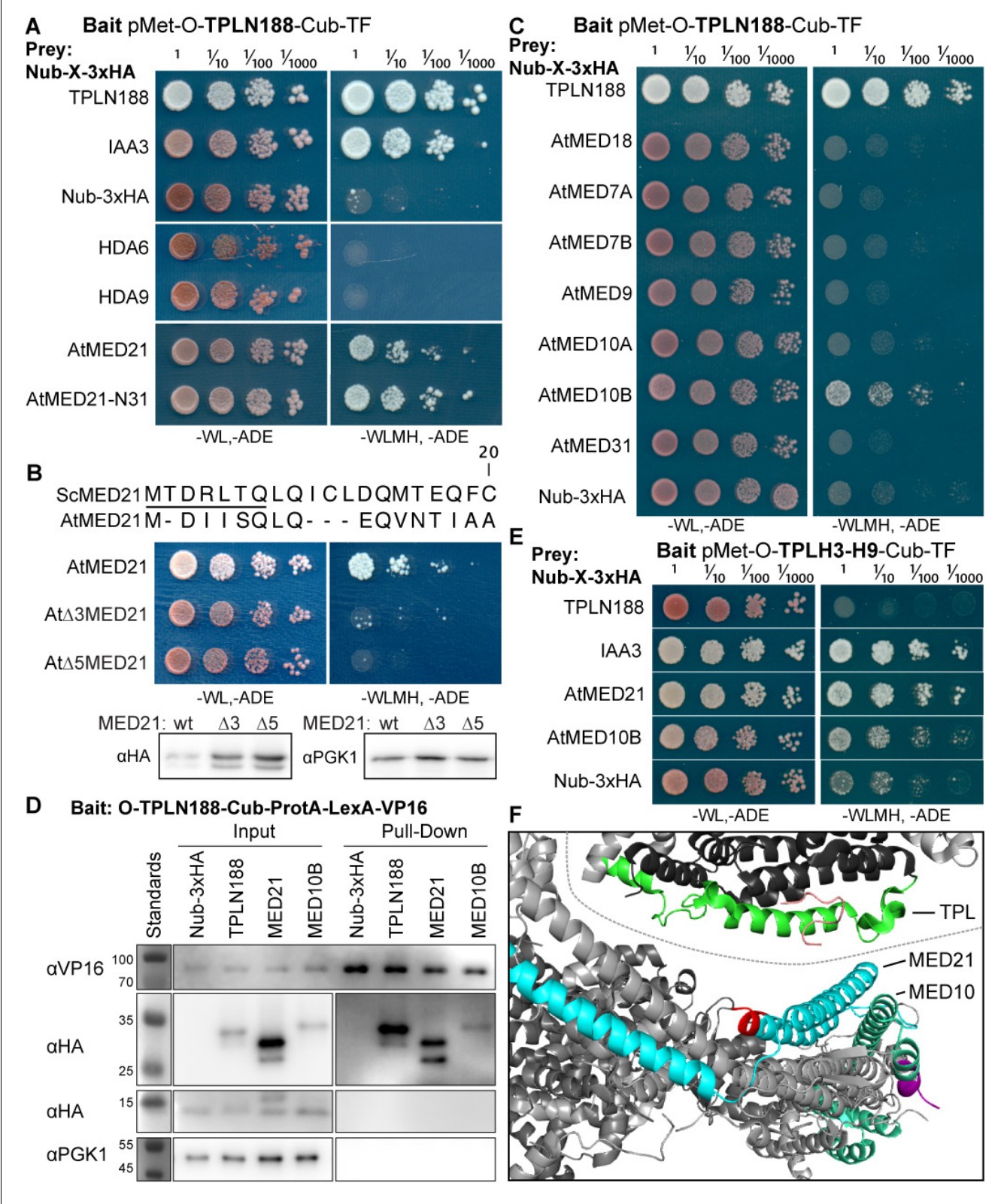

**Figure 2.** The Helix 8 repression domain of TPL directly interacts with AtMED21 and AtMED10B. (A–C, E) Cytoplasmic split-ubiquitin system (CytoSUS) assays with candidate interacting proteins. Nub-3xHA is the N-terminal fragment of ubiquitin expressed with no fusion protein and is used as a negative control. Each prey protein is from *Arabidopsis*. -WL, -ADE: dropout lacking Trp, Leu, and Ade (growth control); -WLMH, -ADE: dropout lacking Trp, Leu, His, Met, and Ade (selective media). The plating for each panel was performed at the same day, white lines are provided when plates were

*Figure 2 continued on next page*

*Figure 2 continued*

cropped for clarity. (**B**) Alignments of the *Arabidopsis* (At) and *Saccharomyces* (Sc) MED21 proteins are shown above cytoSUS assays with the same bait shown in (**A**). Western blots below the colonies indicated that AtMED21 N-terminal Δ3 and Δ5 are well expressed in assay conditions. (**C**) CytoSUS assays with selected Mediator proteins in the middle module. (**D**) The TPL-ProteinA-TF fusion protein can pull down TPL, AtMED21, and AtMED10B from yeast extracts using IgG-beads. Detection of the VP16 transcriptional activator demonstrates enrichment of the fusion protein (αVP16). Each prey protein is detected via the 3xHA tag (αHA), and efficacy of purification was judged by PGK1 depletion (αPGK1). (**E**) A TPL-N truncation lacking the LisH domain (TPLH2-H9) could still interact with the AtMED21-N31 truncation. This bait construct interacted with IAA3, but only minimally with the negative control (free Nub-3xHA). (**F**) Yeast Mediator (bottom, 5N9J) and AtTPL (top, 5NQV) manually juxtaposed to compare relative domain sizes and feasibility of a TPL-MED21-MED10B interaction. TPL Helix 8–9 is colored green. MED21 is colored aqua, with the N-terminus colored red, and the IAA27 EAR peptide in orange. MED10 is colored teal, with the C-terminus colored purple. The dotted line indicates the border between TPL and Mediator structures.

The online version of this article includes the following figure supplement(s) for figure 2:

**Figure supplement 1.** The TPL-N terminal domain (TPLN188) interacts with the N-terminus of AtMED21.

**Figure supplement 2.** Homology and structure of the MED21 subunit of the Mediator complex.

the CRA domain (H3-H8) requires contact with MED21 to repress, and that this is independent of repression via Helix 1.

The large Mediator complex stabilizes the pre-initiation complex (PIC) enabling transcriptional activation (*Kornberg, 2005*; *Nozawa et al., 2017*; *Roeder, 1996*; *Schilbach et al., 2017*). The connection we found between TPL, two Mediator components, and transcriptional repression raised the possibility that other parts of the Mediator complex might also contribute to corepressor function. To test this, we needed a way to measure whether loss of function of individual Mediator components led to de-repression at an individual locus. In the case of the yeast corepressor Tup1, a standard approach has been to test the level of transcription at target genes in the presence of deletion mutations of Tup1-interacting proteins (*Gromöller and Lehming, 2000*; *Lee et al., 2000*; *Zhang and Reese, 2004*). One challenge to such an approach for Mediator components is that loss-of-function mutants can be lethal or exhibit drastic physiological phenotypes (*Biddick and Young, 2005*).

To avoid these complications, we turned to the well-established Anchor Away system for inducible protein depletion (*Haruki et al., 2008*) and combined it with quantification of transcriptional activity at the synthetic locus in the *At*ARC$^S$ (*Figure 4A, B*). However, *At*ARC$^{Sc}$ integrates components at four genomic locations using prototrophic markers that are not compatible with those needed for Anchor Away. To overcome this limitation, we re-created the ARC on a single plasmid (we refer to this plasmid as SPARC) using the Versatile Genetic Assembly System (VEGAS, *Mitchell et al., 2015*). SPARC behaved with similar dynamics to the original *At*ARC$^{Sc}$ on both solid and liquid growth conditions (*Figure 4—figure supplement 1A–C*). As a first test of the Anchor Away system with SPARC, we fused Tup1 and its partner protein Cyc8 to two copies of the FKBP12-rapamycin-binding (FRB) domain of human mTOR (*Haruki et al., 2008*). Rapamycin treatment of strains targeting either of these proteins caused no release of repression on the SPARC reporter, providing confirmation of orthogonality of the synthetic system in yeast (*Figure 4—figure supplement 1D*).

Before testing repressive function, we first performed chromatin immunoprecipitation using the Anchor Away FRB tag to ask whether Mediator proteins or specific Mediator modules were detectable at the promoters of TPL-repressed genes. We began by assaying the integrated AtARC$^{Sc}$ locus and comparing it to a Tup1-repressed locus, *SUC2* (*Carlson and Botstein, 1982*; *Fleming and Pennings, 2007*; *Trumbly, 1992*), and to an active locus enriched for Mediator, *PMA1* (*Petrenko et al., 2017*; *Schmitt et al., 2006*; *Serrano et al., 1986*). We generated a yeast strain with an integrated MED21-FRB fusion protein and an integrated AtARC$^{Sc}$ locus with a 2xHA epitope-tagged TPLN188-IAA3 repressor protein. We observed enrichment of both MED21-FRB and the 2xHA-TPLN188-IAA3 fusion protein at the AtARC$^{Sc}$ locus (*Figure 4C*, *Figure 4—figure supplement 2A–C*). We also observed a modest enrichment of MED21-FRB at the *SUC2* promoter (approximately twofold) and higher enrichment of MED21-FRB at the active *PMA1* promoter (approximately eightfold). Next, we introduced the fully repressed SPARC plasmid containing TPLN188 (SPARC$^{N188}$) into a library of Anchor Away yeast strains that allow specific depletion of Mediator components (see *Figure 4A, B*; *Haruki et al., 2008*; *Petrenko et al., 2017*). We tested representatives of the mediator complex (tail

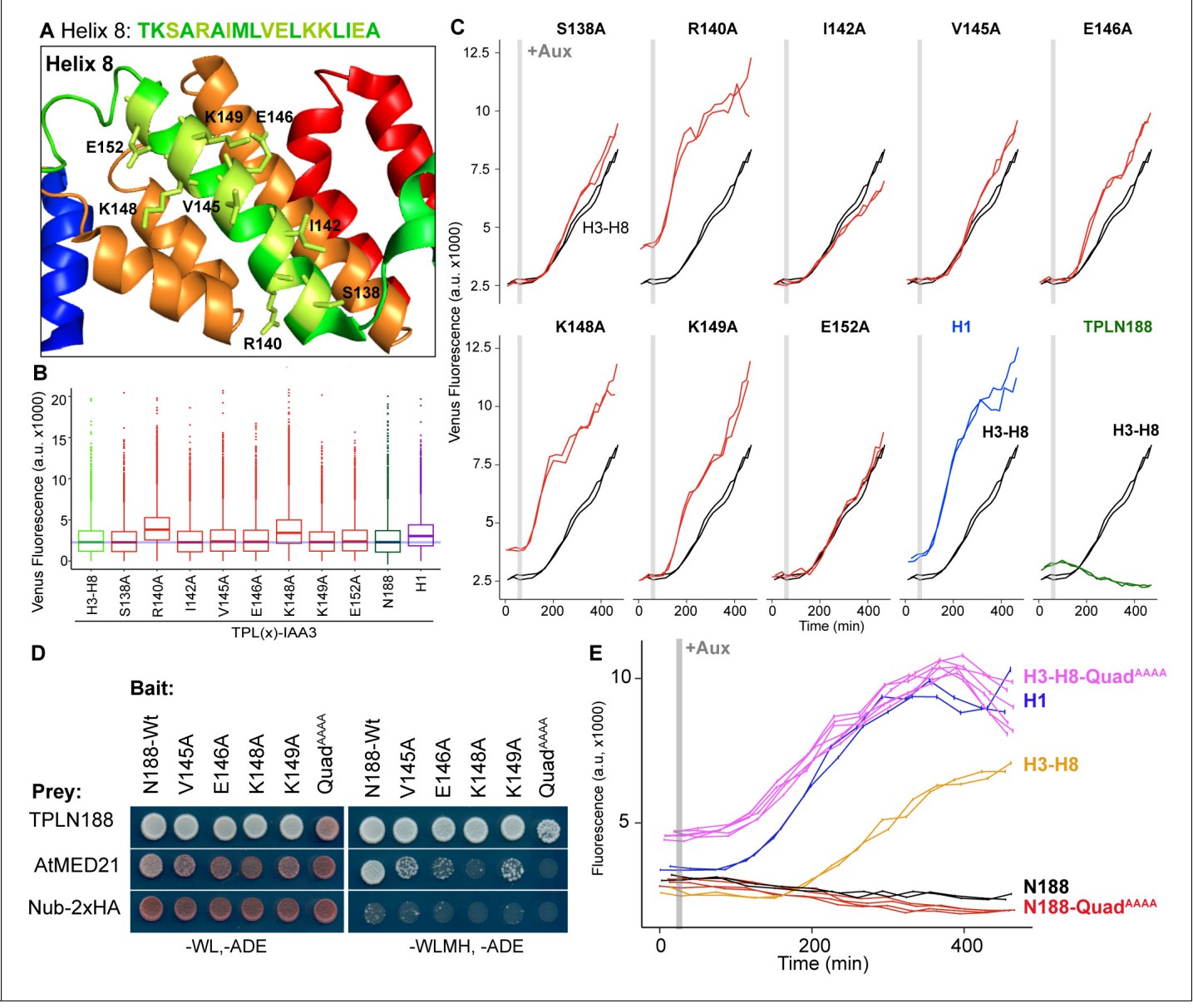

**Figure 3.** Identification of critical residues within Helix 8 repression domain. (**A**) Sequence and structure of Helix 8 (5NQS). Helix 8 is colored green, and amino acids chosen for mutation are highlighted in light green in both the sequence and the structure. (**B**) Repression activity of indicated single- and double-alanine mutations. (**C**) Time-course flow cytometry of selected mutations of Helix 8 following auxin addition. TPLH3-8-IAA3 fusion proteins (black) were compared to indicated single mutations to alanine (red). Controls: Helix 1 (H1 – blue) and TPLN188 (dark green). (**D**) A series of alanine mutations (V145A, E146A, K148A, K149A, and the quadruple mutant Quad$^{AAAA}$ chosen from **A–C**) were introduced into the TPLN188 bait construct and tested for interaction with wild-type TPLN188, AtMED21, and controls. Each single-alanine mutation reduces TPL interaction with AtMED21, while the quad mutation abrogated interaction. (**E**) The Helix 8 Quad$^{AAAA}$ mutation was introduced into the TPLN188-IAA3 and TPLH3-8-IAA3 fusion proteins and compared to wild-type N188 in time-course flow cytometry. For all cytometry experiments, the indicated TPL construct is fused to IAA3. Every point represents the average fluorescence of 5–10,000 individually measured yeast cells (a.u.: arbitrary units). Auxin (IAA-10 µM) was added at the indicated time (gray bar, +Aux). At least two independent experiments are shown for each construct.

– Med15, head – Med18, middle – Med21 and Med14, kinase – CDK8, general transcription factors – TBP1, TFIIA, and RNA Pol II – Rpb1) by ChIP-qPCR. We observed enrichment of all tested core mediator complex members, as well as general transcription factors, at both the SPARC and the *SUC2* loci, with very little enrichment of RNA Pol II (*Figure 4D*). In general, MED21 was detected in lower levels at repressed loci than at the active *PMA1* promoter (*Figure 4D*). Consistent with this observation, Mediator is highly enriched at the SPARC promoter when TPL is absent (*Figure 4—*

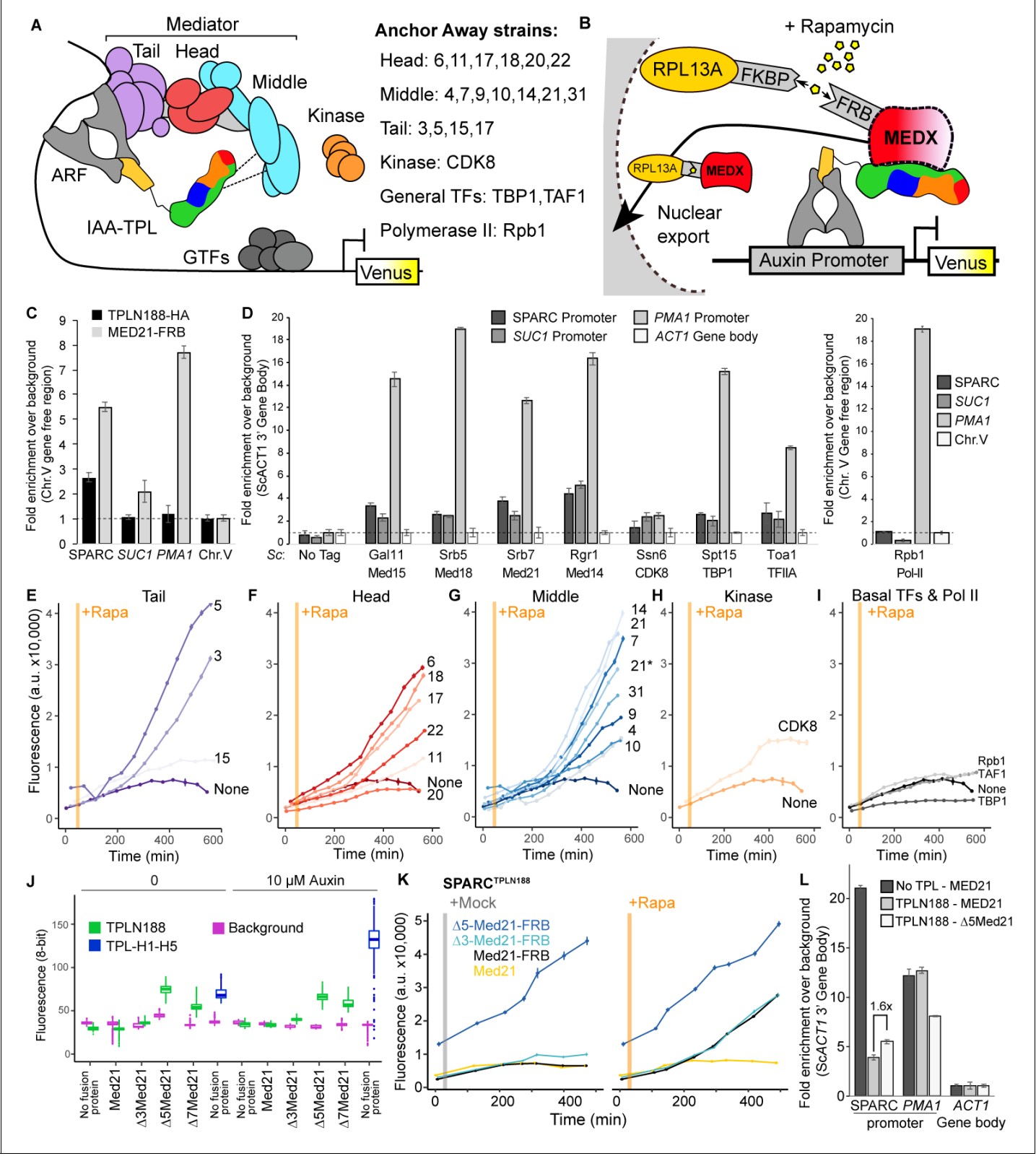

**Figure 4.** Repression by TPL requires interaction with the N-terminus of MED21 at promoters. (**A**) Model of the proposed interaction between the TPL N-terminus with Mediator, where TPL interaction with Mediator 21 and 10 inhibits the recruitment of Pol II. Proteins in this complex that were tested by Anchor Away are listed on the right. (**B**) Schematic of _At_ARC^Sc combined with methods for inducible expression and nuclear depletion of MED21. In Anchor Away, the yeast ribosomal protein 13A (RPL13A) is fused to the rapamycin-binding protein FKBP. Addition of rapamycin induces dimerization

_Figure 4 continued on next page_

*Figure 4 continued*

between FKBP and any target protein fused to 2xFRB, resulting in removal of the target protein from the nucleus. For these experiments, *At*ARC^Sc was assembled into a single plasmid (SPARC) rather than being integrated into separate genomic loci (*Figure 4—figure supplement 1*). (**C**) 2xHA-TPLN188-IAA3 and MED21-FRB association with the ARC and the *ScSUC2* and *ScPMA1* promoters. ChIP was performed with αHA and αFRB before qPCR was used to quantify enrichment at specified loci. (**D**) Association of FRB-tagged components of Mediator and the transcriptional machinery with the SPARC plasmid and the *ScSUC2* promoters. ChIP was performed with αFRB, and qPCR was used to quantify enrichment at specified loci. (**E–I**) Time-course flow cytometry analysis of SPARC^N188 in Mediator Anchor Away yeast strains with rapamycin (orange bar, +Rapa). Two Med21 strains were compared in the Middle domain (**E**), 21 (generated in this study) and 21* (generated in *Petrenko et al., 2017*). Both 21 and 21* demonstrated similar increases in reporter expression. (**J**) Quantification of Venus fluorescence from SPARC^N188 in wild-type and N-terminal ScMed21 deletions with and without auxin. The x-axis indicates strain and which FRB fusion protein is being tested. Yeast was grown for 48 hr on synthetic drop out (SDO) media with or without auxin, and colony fluorescence was quantified and plotted with the auxin-responsive SPARC^H1-H5 in wild type as a reference. Background: red autofluorescence was used as a reference for total cell density. (**K**) Time-course flow cytometry analysis of SPARC^N188 in wild-type and N-terminal ScMed21 deletions with and without rapamycin. Genotypes are indicated in the colored key inset into the graph. For (**E–I, K**) a.u.: arbitrary units. Rapamycin was added at the indicated time (orange bar, +Rapa). Every point represents the average fluorescence of 5–10,000 individually measured yeast cells. (**L**) Association of MED21-FRB or Δ5-MED21-FRB with SPARC plasmids. ChIP was performed with αFRB, and qPCR was used to quantify enrichment at specified loci. (**C, D, L**) A region of the *ACT1* gene body or a gene-free region of chromosome V (Chr.V) was arbitrarily defined as background, and data is presented as fold enrichment over the control gene. Averages and standard errors of four replicates are shown.

The online version of this article includes the following figure supplement(s) for figure 4:

**Figure supplement 1.** Construction and characterization of the single locus auxin response circuit (SPARC).

**Figure supplement 2.** Mediator is detectable at the ARC promoter.

**Figure supplement 3.** N-terminal ScMed21 deletions impair auxin-responsive transcriptional activation.

**Figure supplement 4.** Inducible MED21 rescues rapamycin-induced yeast growth defects.

*figure supplement 2D*). Higher enrichment of members of the middle module at repressed promoters (i.e., Med21, Med14, *Figure 4D*) may point to these subunits nucleating assembly of the entire complex.

We next tested whether the association of Mediator complex components was required for TPL-mediated repression (*Figure 4A, B*). Nuclear depletion of Mediator components from the tail, head, and middle domain triggered clear activation of the SPARC^N188 reporter (*Figure 4E–I*). Depletion of the Mediator kinase module component CDK8 had a more modest effect (*Figure 4H*). One caveat to this approach is that nuclear depletion of components that are absolutely required for transcriptional activation, such as RNA Pol II Anchor Away (Rpb1; *Figure 4I*), cannot be assayed for impacts on repression using transcription of the reporter as the output.

To further interrogate the impact of Mediator on repression, we next focused on the other side of the interaction, namely which region of MED21 was required for interaction with TPL. Deletion of the first seven amino acids of ScMed21 (Δ7Med21) partially activates genes that are normally repressed by Tup1 (*Gromöller and Lehming, 2000*), so we first tested if the same held true for TPL-mediated repression. We introduced SPARCs with different TPL constructs into strains where wild-type ScMed21 or N-terminal deletions were targets of Anchor Away. Importantly, the addition of the FRB tag did not alter ScMED21 function (*Figure 4J*, *Figure 4—figure supplement 1E, F*). We observed that deletion of either five or seven of the N-terminal residues of ScMed21 increased the expression of the reporter in SPARC^N188 to a level similar as what is observed with TPL H1-H5 (*Figure 4J*, *Figure 4—figure supplement 1E, F*). No mutation increased the SPARC's sensitivity to auxin. As Δ7ScMed21 had a noticeable impact on growth, as has been reported previously (*Gromöller and Lehming, 2000*; *Hallberg et al., 2006*), we removed it from further studies. Δ5ScMed21 had no observable growth defects, although this deletion is known to be sufficient to alter Mediator assembly and disrupt binding of Pol II and the CDK8 kinase module (*Hallberg et al., 2006*; *Sato et al., 2016*).

The fully repressed SPARC^N188 in Δ3ScMed21 or Δ5ScMed21 mutants showed elevated reporter transcription when compared to strains carrying wild-type ScMED21 (*Figure 4K*). The addition of rapamycin further increased reporter expression, particularly in the Δ3ScMed21 strains, suggesting that this deletion could only partially disrupt the TPLN188-Med21 interaction. We used chromatin immunoprecipitation to directly test whether Δ5Med21 showed a change in association with the SPARC^N188 promoter. While this deletion would be expected to reduce Med21 association with TPL, the resulting de-repression of the locus should lead to an increase in Mediator association with the activated promoter. Indeed, we observed an ~1.6×-fold increase in Δ5Med21-FRB promoter binding

compared to wild type (*Figure 4L*). While this enrichment was modest compared to a repressor-free SPARC[IAA14] (*Figure 4L*, no TPL; dark gray bar), it was similar to the magnitude of transcriptional activation of the reporter in the Δ5Med21 genotype (*Figure 4J, K*). The well-documented *PMA1* promoter had a substantial enrichment of wild-type MED21, as expected, and was unaffected by the presence of TPL (*Figure 4L*, dark and light gray bars). To confirm that the interaction with TPL was not unique to ScMed21, we replaced the first five amino acids of ScMed21-FRB with the corresponding sequence from AtMED21. The strain carrying this chimeric protein had an identical repression profile as the one with native ScMed21 (*Figure 4—figure supplement 3B*) and showed no difference in growth or viability (*Figure 4—figure supplement 3C*).

To minimize any possible off-target impact of ScMed21 deletions, we introduced estradiol-inducible versions of ScMed21 (iScMed21) into the Anchor Away SPARC[N188] strains (*Figure 4—figure supplement 4*; *McIsaac et al., 2013*). The combination of all three synthetic systems – ARC[Sc], Anchor Away, and estradiol inducibility – made it possible to rapidly deplete the wild-type ScMed21-FRB from the nucleus while simultaneously inducing ScMed21 variants and visualizing the impact on a single auxin-regulated locus. Depletion of nuclear ScMed21 by rapamycin increased levels of the reporter in all genotypes examined (*Figure 4G, K*) while increasing cell size even in short time courses, consistent with its essential role in many core pathways (*Figure 4—figure supplement 4A*; *Gromöller and Lehming, 2000*). When wild-type iScMed21 was induced, there was a rescue of both reporter repression and cell size (*Figure 4—figure supplement 4C, D*), whereas induction of either Δ3 and Δ5 variants resulted in significantly less reporter repression (*Figure 4—figure supplement 4D*). iΔ3Med21 was induced and accumulated at a comparable level to wild-type Med21, while iΔ5Med21 is less stable (*Figure 4—figure supplement 4E, F*). In the time courses with both rapamycin and estradiol, we did not observe the cell size increases observed in the rapamycin treatments alone (populations were evenly distributed around a single mean), suggesting that we were observing the immediate effects of the Med21 deletions (*Figure 4—figure supplement 4G, H*).

Several lines of evidence suggest that, in addition to interactions with other partners, homomultimerization modulates TPL repression potential. First, structures of the N-terminal domains of TPL (*Martin-Arevalillo et al., 2017*) and a rice homolog OsTPR2 (*Ke et al., 2015*) reveal high conservation of residues that coordinate formation of homotetramers and connect tetramer formation to Aux/IAA binding. Second, the dominant TPL mutant *tpl-1* altered a single amino acid in the ninth helix of the TPL-N terminus (N176H) that induces aggregation of TPL and its homologs (TPR1-4), reducing total activity (*Long et al., 2006*; *Ma et al., 2017*). Third, TPL recruitment motifs found in the rice strigolactone signaling repressor D53 induce higher-order oligomerization of the TPL N-terminus, which increases histone binding and transcriptional repression (*Ma et al., 2017*). Our studies in yeast suggest that there may be a more complex relationship between tetramer formation and repression as we have measured strong repressive activity in several constructs that are unlikely (TPLN100; *Pierre-Jerome et al., 2014*) or unable (H1, H3-8; *Figure 1C*) to form tetramers (compare *Figure 1B* with *Figure 5A*). To quantify the potential for interaction among our constructs, we used the cytoSUS assay (*Asseck and Grefen, 2018*). Helix 8 was required for strongest interaction between TPL constructs (*Figure 5—figure supplement 1A*), although this assessment was complicated by the fact that some of the shorter constructs accumulated to significantly lower levels (*Figure 5—figure supplement 1B*). The weak interaction we could observe between full-length TPL-N and the Helix 1 through Helix 3 construct (H1-3) indicated that the TPL LisH domain is sufficient for dimerization. Therefore, while auxin-insensitive repression may require multimeric TPL, this higher-order complex was not required for auxin-sensitive repression mediated by Helix 1 (*Figure 1E*).

To avoid any potential artifacts from analysis of truncated forms of the N-terminus, we next generated site-specific mutations that disrupted multimerization in the context of TPLN188. Martin-Arevalillo et al. had previously identified a quadruple mutation (K102S-T116A-Q117S-E122S) that abrogated the ability of the CRA domain (Helix 6 and Helix 7) to form inter-TPL interactions (*Martin-Arevalillo et al., 2017*). As this mutant form of TPL is only capable of dimerizing through its LisH domain, we refer to it here as LDimer (*Figure 5A*). The LDimer mutations in TPLN188 retained the same auxin-insensitive repression behavior as wild-type TPLN188 (*Figure 5D*), supporting the finding from the deletion series.

To make a fully monomeric form of TPL, we introduced mutations into the dimerization interface of the LisH domain in the context of LDimer. We first mutated one of a pair of interacting residues (F15) to a series of amino acids (tyrosine – Y, alanine – A, arginine – R, or aspartic acid – D) in the

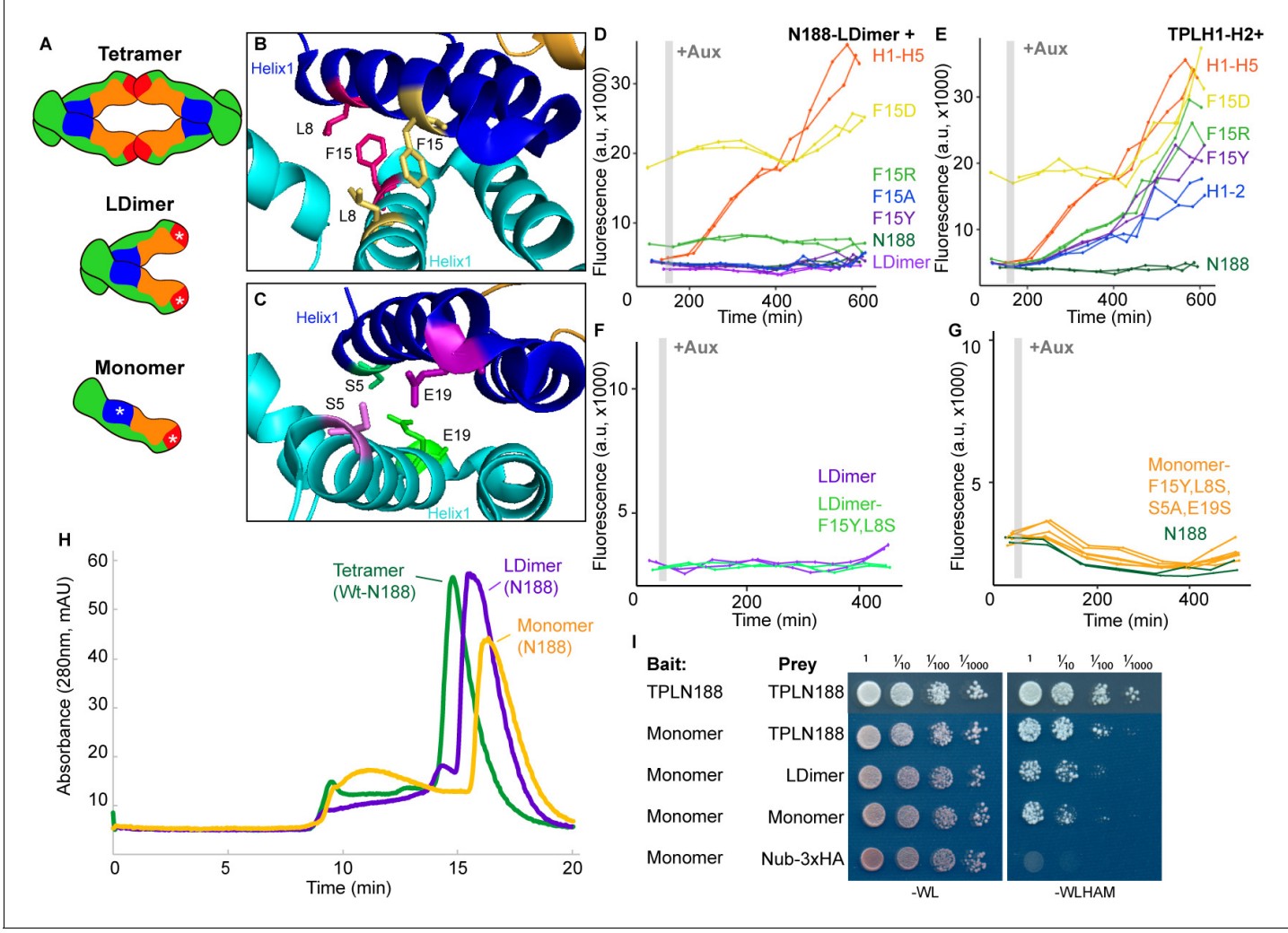

**Figure 5.** Multimerization is not required for repression in yeast. (**A**) TPL can form a homotetramer via the CRA (red) and LisH (blue) domains. Asterisks indicate mutations that block or diminish these interactions. (**B, C**) Locations of critical positions in Helix 1 are highlighted for two interacting TPL monomers (shown in light and dark blue). Interacting amino acids share the same color (adapted from 5NQV). (**D–G**) Time-course flow cytometry analysis of TPLN-IAA3 fusion proteins carrying selected single point mutations in N188-LDimer-IAA3 (**D**) and the TPLH1-2 truncation (**E**). The F15Y mutation had little effect on repression activity for either TPL construct. Double mutations (F15Y, L8S in LDimer) (**F**) or the quadruple Monomer mutations (S5A, L8S, F15Y, E19S in LDimer) (**G**) showed repression activity that was indistinguishable from LDimer or wild-type N188 fused to IAA3. For all cytometry experiments, the indicated TPL construct is fused to IAA3. Every point represents the average fluorescence of 5–10,000 individually measured yeast cells (a.u.: arbitrary units). Auxin (IAA-10 μM) was added at the indicated time (gray bar, +Aux). At least two independent experiments are shown for each construct. (**H**) Size exclusion chromatography on TPLN188 wild-type (green), LDimer (purple), and Monomer (orange) tetramerization mutants. (**I**) Cytoplasmic split-ubiquitin system (CytoSUS) on TPL tetramerization mutants.

The online version of this article includes the following figure supplement(s) for figure 5:

**Figure supplement 1.** TPL multimerization requires Helix 8.

context of either LDimer (*Figure 5D*), or H1-2 (*Figure 5B, E*). Conversion of F15 to the polar and charged aspartic acid (D) completely abolished repression activity, while the positively charged arginine was better tolerated (*Figure 5D, E*). The conversion of F15 to tyrosine had no effect on LDimer (*Figure 5D*), and only a minimal increase in auxin sensitivity in the context of H1-2 (*Figure 5E*). We then combined LDimer-F15Y with a mutation of the coordinating residue L8 to serine with the intention of stabilizing the now solvent-facing residues. The repressive behavior of this mutant was indistinguishable from that of LDimer (*Figure 5F*).

To further push the LDimer towards a monomeric form, we introduced two additional mutations (S5A, E19S, *Figure 5C, G*). Size-exclusion chromatography confirmed that this combination of

mutations (S5A-L8S-F15Y-E19S-K102S-T116A-Q117S-E122S, hereafter called Monomer) successfully shifted the majority of the protein into a monomeric state (*Figure 5H*); however, this shift had no observable impact on repression strength before or after auxin addition (*Figure 5G*). To test whether these mutations had a similar impact on in vivo TPL complexes, we introduced the LDimer and Monomer mutations into the cytoSUS assay. In contrast to the in vitro chromatography results with purified proteins, Monomer expressed in yeast retained measurable interaction with wild-type TPL, LDimer or Monomer, albeit at a reduced level than what was observed between wild-type TPLN188 constructs (*Figure 5I*). A caveat to this apparent difference between assays is that the Monomer mutations led to a striking increase in protein concentration in yeast (*Figure 5—figure supplement 1C*), likely partially compensating for the decrease in affinity.

To ascertain which of our findings about TPL required the sensitivity and simplicity of the synthetic context and which could be observed in the full complexity of intact plant systems, we performed a set of experiments in *Nicotiana benthamiana* (tobacco) and *Arabidopsis.* Bimolecular fluorescence complementation (BiFC) confirmed the interaction between TPL and MED21 (*Figure 6A*), which was further validated by co-immunoprecipitation using tobacco extracts (*Figure 6B*). We were also able to pull down MED21 and TPL using MED10B (*Figure 6—figure supplement 1A*). BiFC also confirmed the importance of the same TPL Helix 8 residues for the TPL-AtMed21 interaction (*Figure 6A*, TPL[H8QuadA]). Similarly, the Δ5AtMED21 N-terminal truncation eliminated interaction with full-length TPL (*Figure 6A*). We next developed a quantitative repression assay based on UAS/GAL4-VP16 (*Brand and Perrimon, 1993*; *Figure 6C*). To block potentially confounding interactions with endogenous TPL/TPRs or TIR1/AFBs, we engineered a variant of IAA14 with mutations in the two EAR domains (EAR[AAA]) and in the degron (P306S) (IAA14[mED]; *Figure 6C*). After prototyping the system in yeast (*Figure 6—figure supplement 1B, C*), we quantified repression strength of constructs carrying *TPLN-IAA14[mED]* variants using the well-characterized synthetic auxin-responsive promoter DR5 (*Ulmasov et al., 1997*). As expected, DR5 was strongly induced by co-transformation with AtARF19, and this induction was sharply reduced by the inclusion of *UAS-TPLN188-IAA14[mED]* and *GAL4-VP16* (*Figure 6—figure supplement 1D*). Overall, we observed strong correlation in repression activity between what was observed in yeast and in tobacco.

To connect the observed differences in repression strength to a developmental context, we generated transgenic *Arabidopsis* lines where the UAS-TPL-IAA14[mED] constructs were activated in the cells where IAA14 normally acts to regulate the initiation of lateral root primordia (*Figure 6D*; *Gala et al., 2021*; *Laplaze et al., 2005*). Expression of functional TPL-IAA14[mED] fusion proteins in these xylem pole pericycle cells should strongly suppress production of lateral roots, phenocopying the solitary root (*slr*) mutant, which carries an auxin-resistant form of IAA14 (*Fukaki et al., 2002*). Indeed, TPLN188 fusion constructs sharply decreased lateral root density (*Figure 6E*), while transformants expressing either IAA14[mED] (with no TPL fusion) or TPLN188 (with no IAA14 fusion) had no effect on lateral root production (*Figure 6E*). TPLH3-H9 decreased lateral root density albeit not as effectively as TPLN188, suggesting that Helix 1 is required for full repression in a native context (*Figure 6E*). Both LDimer and Monomer constructs (*Figure 6E*) were able to repress lateral root development to the same extent as TPLN188, meaning that multimer formation is not required for TPL-mediated repression in this context. The fusion containing the Helix 8 quadruple mutant demonstrated a clear loss of repression, indicating that the TPL-MED21 interaction is critical for repression when expressed in lateral root-forming cells (*Figure 6E*).

Given this result, we wanted to directly test the role of AtMed21 in auxin-regulated development in the presence of native isoforms of TPL and Aux/IAAs. This was complicated by the fact that, as in yeast, AtMED21 is essential in plants. While homozygous loss-of-function mutations are embryo lethal (*Dhawan et al., 2009*), plants heterozygous for *Atmed21* mutations appear wild type (*Figure 6—figure supplement 2A, B*). To overcome this obstacle, we took two approaches that relied on the same xylem pole pericycle driver as described for the TPL functional assays. First, we expressed N-terminal deletion variants of MED21 that should weaken or sever interaction with TPL (*Figure 6F*; Δ3MED21, Δ5MED21, Δ7MED21). If MED21-TPL interaction is critical to maintain normal expression of the lateral root program, reduced interaction should trigger an increase in lateral root density. This was exactly what we observed for transgenic lines expressing any of the three deletions. Second, we repressed transcription of *AtMED21* by introducing a dCAS9-TPLN300 synthetic repressor under the control of a UAS promoter along with three sgRNAs complementary to the *AtMED21* promoter. Similar to the predictions above, reduced expression of AtMED21 in xylem

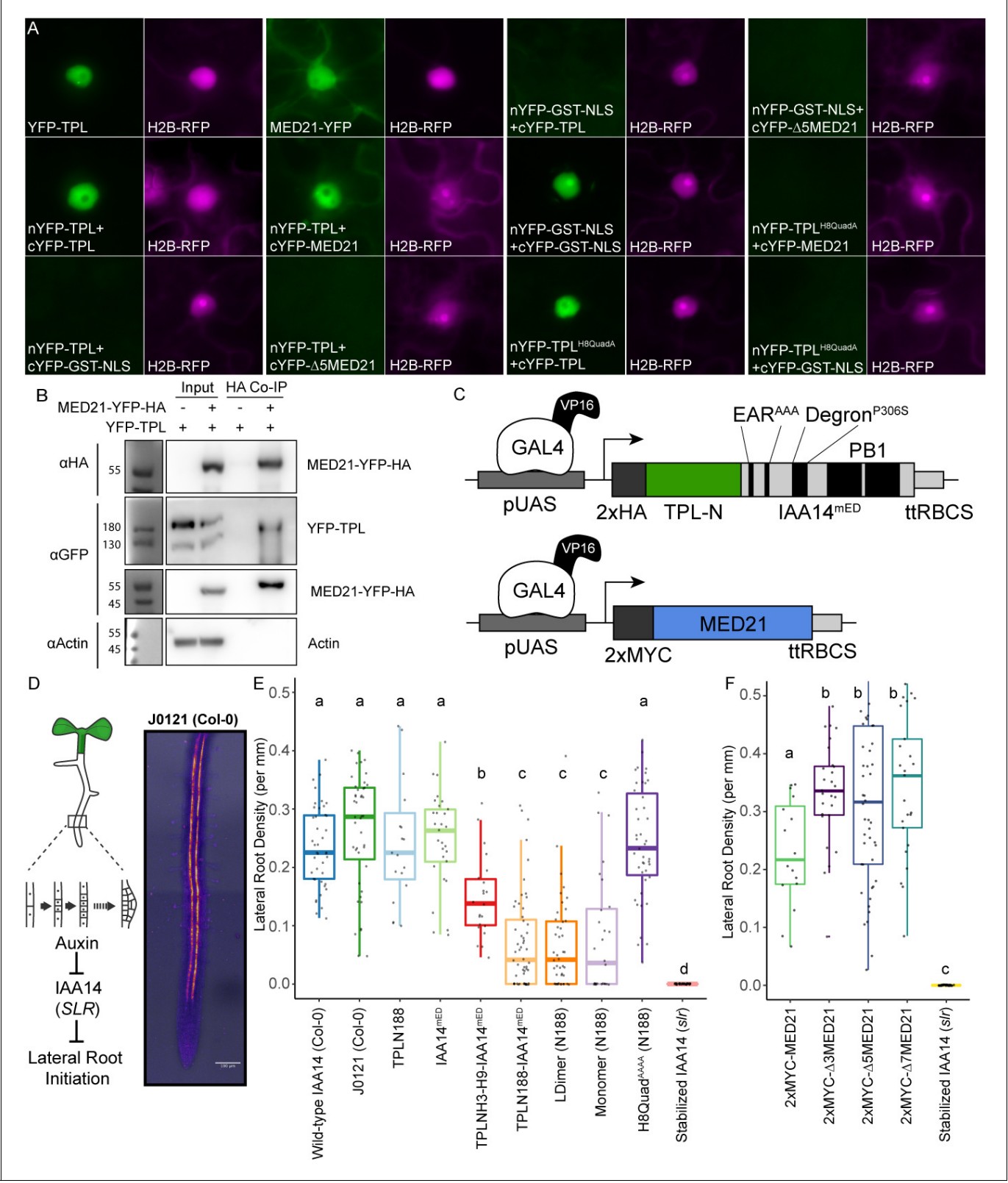

**Figure 6.** The TPL CRA repression domain behaves similarly in yeast and plants. (**A**) Bimolecular fluorescence complementation assay performed in tobacco. Each image is an epi-fluorescent micrograph taken at identical magnification from tobacco epidermal cells at 2 days post injection. The YFP image is colored green (left panel). p35S:H2B-RFP was used as a control and is false-colored magenta (right panel). (**B**) Co-immunoprecipitation of MED21 and TPL from tobacco leaves. MED21-YFP-HA was immunoprecipitated using anti-HA, and YFP-TPL was detected using the YFP fusion. Actin

*Figure 6 continued on next page*

*Figure 6 continued*

was used to demonstrate that the purification had removed non-specific proteins. Numbers on the left of blots indicate sizes of protein standards in kilodaltons. (C) Design of UAS-TPL-IAA14^mED and UAS-MED21 constructs. Mutation of the conserved lysine residues in the EAR domain disrupted potential interactions with endogenous TPL/TPR proteins. The IAA14 degron has been mutated (P306S) to render it auxin insensitive. UAS: upstream activating sequence; ttRBCS: Rubisco terminator sequence. (D) Auxin-induced degradation of IAA14 is absolutely required for initiation of lateral root development (cartoon, left). An enhancer trap line (J0121) expresses GAL4-VP16 and UAS-GFP in in xylem pole pericycle cells. (E) N-terminal domains of TPL were sufficient to repress the development of lateral roots in *Arabidopsis* seedlings. The density of emerged lateral roots was measured in T1 seedlings at 14 days after germination. (F) N-terminal deletions in *AtMED21* were sufficient to dominantly increase the development of lateral roots in *Arabidopsis* seedlings. The density of emerged lateral roots was measured in T1 seedlings at 14 days after germination. (E, F) Lowercase letters indicate significant difference (ANOVA and Tukey HSD multiple comparison test; p<0.001).

The online version of this article includes the following figure supplement(s) for figure 6:

**Figure supplement 1.** The TPL-MED21 interaction is required for repression in plants.

**Figure supplement 2.** The essential gene MED21 is required for normal lateral root development in plants.

pole pericycle cells should stimulate lateral root development. This is indeed what we observed (*Figure 6—figure supplement 2C–E*).

Previous reports connected the Mediator kinase module, and specifically the function of AtMED13/GRAND CENTRAL (GCT), to repression by TPL (*Ito et al., 2016*). We did not find evidence for this relationship in our yeast assays and wanted to investigate further. Although embryo lethality is seen in the majority of homozygous *med13/gct* individuals, a small percentage survive. This made it possible to examine the effect of expressing wild-type and monomeric forms of TPLN188 in xylem pole pericycle cells in the absence of MED13/GCT activity (*Figure 6—figure supplement 1E, F*). Loss of *med13* function was unable to rescue the production of lateral roots in these lines, leading to the conclusion that TPLN188 requires AtMED21 but not AtMED13 for repression, at least in this context.

## Discussion

A review of the current literature on corepressors gives the conflicting impressions that (a) corepressor function is broadly conserved and (b) that every organism (and perhaps even every corepressor) has a distinct mode for transcriptional repression (*Adams et al., 2018*; *Mottis et al., 2013*; *Perissi et al., 2010*; *Wong and Struhl, 2011*). We hoped that the *At*ARC^Sc could facilitate a resolution to this apparent contradiction by targeting repression to a single synthetic locus. We focused our initial efforts on the analysis of the N-terminal portion of TPL, which has multiple known protein-protein interaction surfaces (*Ke et al., 2015*; *Martin-Arevalillo et al., 2017*). Experiments with the *At*ARC^Sc identified two independent repression domains (*Figure 1*), and we focused additional study on the stronger of the two that was localized to the CRA domain. Within this domain, we were able to identify two interacting partners that are part of the middle module of the Mediator complex: MED21 and MED10 (*Figure 2*). Four amino acids within Helix 8 with R-groups oriented away from the hydrophobic core of the TPL structure were found to be required for both Med21-binding and repressive function (*Figure 3*). Indeed, the entire core Mediator complex (head, middle, tail) appears to be recruited to TPL-repressed loci and required to maintain repression (*Figure 4*). Contrary to our initial hypothesis, the monomeric form of TPL was sufficient for strong repression in yeast and in plants (*Figure 5*), leaving open the question of the role of higher-order TPL complex formation. Finally, we were able to confirm that our insights from synthetic assays in yeast were relevant to regulation of the auxin-mediated lateral root development pathway in intact plants (*Figure 6*).

Corepressors coordinate multiple mechanisms of repression through discrete protein interactions, leading to robust control over eukaryotic transcription by combining repression modalities. Corepressor function has variously been linked to (a) altering chromatin confirmation, often through interaction with histone-modifying proteins or histone proteins themselves, (b) direct interference with transcription factor binding or function, and (c) physical spreading of long-range oligomeric corepressor complexes across regions of regulatory DNA (*Perissi et al., 2010*). Dissection of the importance of each modality in Tup1 repression has been challenging (*Lee et al., 2000*; *Zhang and Reese, 2004*). The tour-de-force of corepressor mechanism studies in yeast concluded that the primary function of Tup1 was to physically block activators (*Wong and Struhl, 2011*). In their work, the

authors utilized the Anchor Away approach to correlate the importance of HDACs, transcriptional machinery, and chromatin remodeling enzymes to the repression state of endogenously repressed Cyc8-Tup1 target genes. They observed that Tup1 did not block the binding of transcription factors but inhibited the recruitment of one Mediator component in the tail domain, GAL11/MED15, as well as Pol II and the chromatin remodelers Snf2 and Sth1. They additionally observed that HDACs had only a supportive role in reinforcing Tup1 repression. These results led to their hypothesis that Tup1 blocks the activation domains of transcription factors and suggested this was through direct binding to activation domains (*Wong and Struhl, 2011*).

The synthetic system used here allowed us to build on this model and further refine our understanding of TPL's repressive activity. In our experiments, we see a similar set of conditions, with TPL recruited to the DNA-bound transcriptional activator (ARF), and several possible mechanisms of repression. Unlike Tup1, we have subdivided the TPL protein to identify interactions between TPL and individual protein interactors with no effect on yeast function. In these experiments, we can eliminate the possibility that TPL blocks ARF activation by directly blocking the transcription factor activation domains because we see a loss of repression only when TPL-MED21 binding is eliminated through specific point mutations (*Figure 4J–L*). Our estradiol-inducible replacement assays where different isoforms of Med21 are expressed also corroborate these findings (*Figure 4—figure supplement 4*) as the SPARC remains genetically identical in these strains, indicating that TPL-MED21 interaction is regulating Mediator activity not a TPL-ARF interaction. Furthermore, our results correlate well with findings that repressed targets are reactivated when this portion of MED21 is deleted in yeast (TPL, *Figure 4*; Tup1; *Gromöller and Lehming, 2000*). Therefore, we suggest that instead of directly binding activation domains that TPL (and likely Tup1) binds to components of Mediator (MED21, MED10B, and possibly others) recruited by the transcription factor. Indeed, it is easier to rationalize that the repressor binds the same domains of the Mediator complex recruited by the transcription factor's activation domain (with the same affinity) as opposed to binding each diverse activation domain (with varying affinity). In this model, corepressor binding blocks formation of a fully active Mediator complex, thereby limiting Pol II recruitment and promoter escape (*Petrenko et al., 2017*).

The Mediator complex is a multi-subunit complex that connects DNA-bound transcription factors and the RNA Pol II complex to coordinate gene expression (*Flanagan et al., 1991*; *Kim et al., 1994*; *Kornberg, 2005*). The yeast Mediator subunits are organized into four separate modules, head, middle, tail, and kinase, with a strong conservation of module components in plants (*Dolan and Chapple, 2017*; *Maji et al., 2019*; *Malik et al., 2017*; *Samanta and Thakur, 2015*). Med21 forms a heterodimer with Med7 and interacts with Med10, among others, to create the central region of the middle region of the Mediator complex. The Med21 N-terminus is centered on a flexible hinge region (*Baumli et al., 2005*), which is required for recruitment of Pol II and the CDK8 kinase module (*Sato et al., 2016*). The protein interaction between TPL and MED21 occurs at the N-terminus of MED21, highlighting the importance of this region as a signaling hub (*Sato et al., 2016*). Other lines of evidence support this role as this region binds the yeast homolog of TPL, Tup1 (*Gromöller and Lehming, 2000*), through a completely different protein domain as no homology can be found between TPL Helix 8 and Tup1 in any region by primary amino acid homology (i.e., BLAST).

As suggested by Ito and colleagues (*Ito et al., 2016*) and supported by our synthetic system, auxin-induced removal of TPL is sufficient to induce changes in the activity of the Mediator complex; however, multiple points of contact likely exist between the Mediator complex and other parts of the transcriptional machinery in both transcriptionally repressed and active states. For auxin response, specifically, there are several lines of evidence to support this model, including documented association between the structural backbone of Mediator, MED14, and activated and repressed auxin loci in *Arabidopsis* (*Ito et al., 2016*). In addition, MED12 and MED13 are required for auxin-responsive gene expression in the root, and MED12 acts upstream of AUX1 in the root growth response to sugar (*Raya-González et al., 2018*). MED18 in the head module represses auxin signaling and positively regulates the viability of the root meristem (*Raya-González et al., 2018*). PFT1/MED25 regulates auxin transport and response in the root (*Raya-González et al., 2014*). MED7, MED21's partner protein in the hinge domain, is required for normal root development, and loss of MED7 function impacts expression of auxin signaling components (*Kumar et al., 2018*). Previous research identified the Mediator CDK8 module, specifically MED13 (MAB2), as an interactor with the full-length TPL protein (*Ito et al., 2016*). We could not observe interaction between the

N-terminal domain of TPL and AtMED13, AtCYC8, or AtCYCC (*Figure 2—figure supplement 1B*), suggesting that any direct interactions occur outside the N-terminal region.

The conserved interaction of both TPL and Tup1 with Mediator has implications for modeling eukaryotic transcription (e.g., *Estrada et al., 2016*). By stabilizing the Mediator complex, TPL (and by extension Tup1) may create a 'pre-paused' state that allows rapid recruitment of Pol II and activation once TPL is removed. This would be compatible with the multiple repression mechanisms described for TPL at different genetic loci. TPL recruitment of the repressive CDK8 Mediator complex (*Ito et al., 2016*), chromatin remodeling enzymes such as HD19 (*Long et al., 2006*), and contact with histone proteins (*Ma et al., 2017*) would be removed with TPL upon relief of repression. It will be critical in the future to understand how these various forms of repression interact, and especially to map the dynamics of assembly and disassembly of complexes as loci transition from repressed to active states and back to repressed once again.

## Code availability statement

All codes are available through Github: https://github.com/achillobator/TPL_Structure_Function/ (*Leydon, 2021* copy archived at swh:1:rev:141d7d05fe0c23be55af5050563d160f019d6d65).

# Materials and methods

**Key resources table**

| Reagent type (species) or resource | Designation | Source or reference | Identifiers | Additional information |
|---|---|---|---|---|
| Gene (*Arabidopsis thaliana*) | TOPLESS, TPL | GenBank | AT1G15750 | |
| Gene (*Arabidopsis thaliana*) | MEDIATOR 21, MED21 | GenBank | AT4G04780 | |
| Strain, strain background (*Saccharomyces cerevisiae*) | Anchor Away strains | EURO- SCARF euroscarf.de | HHY168 | See Yeast Strain list (*Supplementary file 3*) |
| Strain, strain background (*Saccharomyces cerevisiae*) | cytoSUS strains | *Asseck and Grefen, 2018* | THY.AP4, THY.AP5 | See Yeast Strain list (*Supplementary file 3*) |
| Strain, strain background (*Escherichia coli*) | Rosetta 2 strain | Sigma-Aldrich | 71400 | Electrocompetent cells |
| Strain, strain background (*Nicotiana benthamiana*) | *Nicotiana benthamiana* (wild-type) | GenBank | NCBI: txid4100 | |
| Strain, strain background (*Agrobacterium tumefaciens*) | GV3101 | GenBank | NCBI: txid358 | Electrocompetent cells |
| Genetic reagent (*Arabidopsis thaliana*) | J0121 (in *Col-0* accession) | *Gala et al., 2021* | J0121 | |
| Genetic reagent (*Arabidopsis thaliana*) | *slr* | TAIR | *SLR-1*, AT4G14550 | |
| Genetic reagent (*Arabidopsis thaliana*) | *med21-1* | Arabidopsis Biological Resource Center | WiscDsLox461-464K13 | |

*Continued on next page*

*Continued*

| Reagent type (species) or resource | Designation | Source or reference | Identifiers | Additional information |
|---|---|---|---|---|
| Genetic reagent (*Arabidopsis thaliana*) | *med21*$^{i214G}$ | This paper | | |
| Antibody | Anti-HA-HRP (Rat Monoclonal) | Roche/Millipore Sigma | RRID:AB_390917, REF-12013819001, Clone 3F10 | WB (1:1000) |
| Antibody | Anti-FLAG (Mouse Monoclonal) | Millipore Sigma | RRID:AB_259529, F3165 | WB (1:5000) |
| Antibody | Anti-FRB (Rabbit Polyclonal) | Enzo Life Sciences, (*Haruki et al., 2008*) | RRID:AB_2051920, ALX-215-065-1 | WB (1:10,000) |
| Antibody | Anti-VP16 (1-21) (Mouse Monoclonal) | Santa Cruz Biotechnology | RRID:AB_628443, sc-7545 | WB (1:5000) |
| Antibody | Anti-GFP (Rabbit Polyclonal) | AbCam | RRID:AB_303395, ab290 | WB (1:10,000) |
| Antibody | Anti-MYC (Rabbit Monoclonal) | Cell Signaling | RRID:AB_490778, 71d10, 2278S | WB (1:5000) |
| Antibody | Anti-PGK1 (Mouse Monoclonal) | AbCam | RRID:AB_10861977, ab113687 | WB (1:10,000) |
| Recombinant DNA reagent | RPL13A-FKBP fusion proteins | *Haruki et al., 2008* | | See Plasmid list (*Supplementary file 2*) |
| Peptide, recombinant protein | TPL-6xH | This paper | TPL-6xHis tagged fusion proteins | |
| Commercial assay or kit | DNA Sequencing | Genewiz | Genewiz.com | |
| Chemical compound, drug | Rapamycin | LC Laboratories | R-5000 | 1 µM for Anchor Away |
| Chemical compound, drug | β-estradiol | Sigma | E2758-1G | |
| Chemical compound, drug | Auxin | plantMedia, plantmedia.com | CAT#705490 | (IAA-10 µM) |
| Chemical compound, drug | Geneticin | Thermo Fisher Scientific | G418 | |
| Software, algorithm | CLC Sequence Viewer 7 | QIAGEN | | |
| Software, algorithm | R | R Studio | rstudio.com/ | |
| Software, algorithm | ImageJ | *Schneider et al., 2012* | https://imagej.nih.gov/ij/ | |
| Software, algorithm | SmartRoot | Jülich Research Centre and ROot and Soil/Shoot Interactions virtual group | https://smartroot.github.io/ | |
| Software, algorithm | NeuronJ | Erik Meijering | https://imagescience.org/meijering/software/neuronj/manual/ | |

## Cloning

Construction of TPL-IAA3 and TPL-IAA14 fusion proteins was performed by Golden Gate cloning as described in *Pierre-Jerome et al., 2014*. Variant and deletion constructs were created using PCR-mediated site-directed mutagenesis. Site-directed mutagenesis primers were designed using NEBasechanger and implemented through Q5 Site-Directed Mutagenesis (NEB, Cat #E0554S). TPL interactor genes were amplified as cDNAs from wild-type Col-0 RNA using reverse transcriptase (SuperScript IV Reverse Transcriptase, Invitrogen) and gene-specific primers from IDT (Coralville, IA), followed by amplification with Q5 polymerase (NEB). These cDNAs were subsequently cloned into plasmids for cytoSUS using a Gibson approach (*Gibson et al., 2009*) through the Aquarium Biofabrication facility (*Ben Keller et al., 2019*). The coding sequence of the genes of interest was confirmed by sequencing (Genewiz; South Plainfield, NJ). For UAS-driven constructs, the TPLN188-IAA14 coding sequence was amplified with primers containing engineered BsaI sites and introduced into the pGII backbone with the UAS promoter and RBSC terminator (*Siligato et al., 2016*) using Golden Gate cloning (*Weber et al., 2011*). Subsequent mutations were performed on this backbone using PCR-mediated site-directed mutagenesis (see above). Construction of C-terminal 2xFRB fusions for Anchor Away was done as described in *Haruki et al., 2008*. Inducible MED21 was constructed as described in *McIsaac et al., 2013*. For cell type-specific knockdown mediated by dCas9-TPLN300, Gibson cloning was used to modify the pHEE401E plasmid, replacing the egg-specific promoter and Cas9 from pHEE401E (*Wang et al., 2015*) with the GAL4-UAS promoter and dCas9-TPLN300 fusion protein (*Khakhar et al., 2018*). The resulting plasmid is used as starting point to clone three sgRNAs targeting the *AtMED21* promoter (identified using CHOP-CHOP *Labun et al., 2019*). (sgRNAs: GACGCAGAGTCTGTTGGGTGTGG, TTTAAAATGGGCTTTTAAGGTGG, AACACTGAAGTAGAA TTGGGTGG ranging from −170 to +90 region from the TSS) using PCR and Golden Gate cloning strategy described in *Wang et al., 2015*.

## Flow cytometry

Fluorescence measurements were taken using a Becton Dickinson (BD) special order cytometer with a 514 nm laser exciting fluorescence that is cut off at 525 nm prior to photomultiplier tube collection (BD, Franklin Lakes, NJ). Events were annotated, subset to singlet yeast using the FlowTime R package (*Wright et al., 2019*). A total of 10,000–20,000 events above a 400,000 FSC-H threshold (to exclude debris) were collected for each sample and data exported as FCS 3.0 files for processing using the flowCore R software package and custom R scripts (*Supplementary file 1*; *Havens et al., 2012*; *Pierre-Jerome et al., 2017*). Data from at least two independent replicates were combined and plotted in R (ggplots2).

## Yeast methods

Standard yeast drop-out and yeast extract–peptone–dextrose plus adenine (YPAD) media were used, with care taken to use the same batch of synthetic complete (SC) media for related experiments. A standard lithium acetate protocol (*Gietz and Woods, 2002*) was used for transformations of digested plasmids. All cultures were grown at 30°C with shaking at 220 rpm. Anchor Away approaches were followed as described in *Haruki et al., 2008*, and Anchor Away strains were obtained from EURO-SCARF (euroscarf.de). Endogenous genomic fusions of ScMed21-FRB were designed by fusing MED21 homology to the pFA6a-FRB-KanMX6 plasmid for chromosomal integration into the parental Anchor Away strain as in *Petrenko et al., 2017*, selectable through G418 resistance (G418, Geneticin, Thermo Fisher Scientific). Tup1-FRB and Cyc8-FRB were constructed as described in *Wong and Struhl, 2011*. Mediator Anchor Away strains were created in *Petrenko et al., 2017* and kindly donated by Dr. Kevin Struhl. SPARC construction required a redesign of promoters and terminators used in the *At*ARC^Sc to eliminate any repetitive DNA sequences (see *Figure 4—figure supplement 1*), using a Golden Gate cloning approach into level 1 vectors. Subsequent assembly of individual transcriptional units into a larger plasmid utilized VEGAS assembly, which was performed as described in *Mitchell et al., 2015*. To create an acceptor plasmid for the assembled transcriptional units, we synthesized a custom vector containing VA1 and VA2 homology sites for recombination (Twist Bioscience, South San Francisco, CA). In between these sites, we incorporated a pLac:mRFP cassette to allow identification of uncut destination plasmid in *Escherichia coli*, flanked by EcoRI sites for linearization. Finally, the CEN6/ARSH4 was transferred from pRG215

(Addgene #64525) into the acceptor plasmid by Golden Gate reaction using designed BsmBI sites engineered into the acceptor plasmid and the primers used to amplify the CEN/ARS (see *Figure 4— figure supplement 1*). For the cytoplasmic split-ubiquitin protein-protein interaction system, bait and prey constructs were created using the plasmids pMetOYC and pNX32, respectively (Addgene, https://www.addgene.org/Christopher_Grefen/). Interaction between bait and prey proteins was evaluated using a modified version of the split ubiquitin technique (*Asseck and Grefen, 2018*). After 2 days of growth on control and selection plates, images were taken using a flatbed scanner (Epson America, Long Beach, CA). Inducible ScMed21 strains (iMed21) were grown overnight, and then diluted back to 100 events per microliter as determined by flow cytometry and grown at 30°C with 250 rpm in a deepwell 96-well plate format. Strains were supplemented with β-estradiol (20 μM) for 4 hr followed by rapamycin addition. Samples were analyzed by flow cytometry throughout these growth experiments.

## Western blot

Yeast cultures that had been incubated overnight in SC media were diluted to $OD_{600}$ = 0.6 and incubated until cultures reached $OD_{600}$ ~1. Cells were harvested by centrifugation. Cells were lysed by vortexing for 5 min in the presence of 200 μl of 0.5 mm diameter acid washed glass beads and 200 μl SUMEB buffer (1% SDS, 8 M urea, 10 mM MOPS, pH 6.8, 10 mM EDTA, 0.01% bromophenol blue, 1 mM PMSF) per 1 OD unit of original culture. Lysates were then incubated at 65° for 10 min and cleared by centrifugation prior to electrophoresis and blotting. Antibodies: anti-HA-HRP (REF-12013819001, Clone 3F10, Roche/Millipore Sigma, St. Louis, MO), anti-FLAG (F3165, Monoclonal ANTI-FLAG M2, Millipore Sigma, St. Louis, MO), anti-FRB (ALX-215-065-1, Enzo Life Sciences, Farmingdale, NY; *Haruki et al., 2008*), anti-VP16 (1-21) (sc-7545, Santa Cruz Biotechnology, Dallas TX), anti-GFP (ab290, AbCam, Cambridge, UK), anti-MYC (71d10, 2278S, Cell Signaling, Danvers, MA), and anti-PGK1 (ab113687, AbCam).

## Protein expression and purification

All multimer-deficient TPL proteins were expressed in *E. coli* Rosetta 2 strain. Bacteria cultures were grown at 37°C until they achieved an $OD_{600}$nm of 0.6–0.9. Protein expression was induced with isopropyl-β-D-1-thyogalactopiranoside (IPTG) at a final concentration of 400 μM at 18°C overnight. Bacteria cultures were centrifuged and the pellets were resuspended in the buffer A (CAPS 200 mM pH 10.5, NaCl 500 mM, TCEP 1 mM), where cells were lysed by sonication. His-tagged AtTPL188 (wt and mutants) bacteria pellets were resuspended in buffer A with EDTA-free antiprotease (Roche). The soluble fractions recovered after sonication were passed through a Ni-sepharose (GE Healthcare) column previously washed with buffer A, and the bound proteins were eluted with buffer A with 250 mM imidazole. A second purification step was carried out on Gel filtration Superdex 200 10/300 GL (GE Healthcare) equilibrated with buffer A.

## Co-immunoprecipitation

Co-IP from yeast was performed using the cytoSUS strains. Cultures were grown to $OD_{600}$ 0.5 (~1E7 cells/ml) using selective media, harvested, and resuspended in 200 μl extraction buffer (1% SDS, 10 mM MOPS, pH 6.8, 10 mM EDTA) with protease inhibitors. Cells were lysed by vortexing 3 × 1 min full speed with 100 μl of 0.5 mm Acid Washed Glass Beads, clarified by centrifugation (1 min, 1000 rpm), and supernatant was mixed with 1 ml IP buffer (15 mM $Na_2HPO_4$, mw 142; 150 mM NaCl, mw 58; 2% Triton X-100, 0.1% SDS, 0.5% DOC, 10 mM EDTA, 0.02% $NaN_3$) with protease inhibitors and incubated with 100 μl of IgG sepharose at 25°C for 2 hr with rotation. The beads were washed 1× with IP buffer and 2× with IP-wash buffer (50 mM NaCl, mw58; 10 mM TRIS, mw 121; 0.02% $NaN_3$) with protease inhibitors. Protein was eluted with 50 μl of SUME (1% SDS, 8 M urea, 10 mM MOPS, pH 6.8, 10 mM EDTA) buffer +0.005% bromophenol blue by incubation at 65°C for 10 min and run on handmade 12% acrylamide SDS-PAGE gels, and western blotted accordingly. Co-IPs from tobacco were performed on leaves 2 days after injection as described in *Song et al., 2014* using 35S:MED21-YFP-HA, 35S:TPL-YFP (full length), and 35S:MED10B-MEC-ProtA constructs. For co-IPs with HA, extracts were incubated with Anti-HA-Biotin (High Affinity [3F10], Sigma, 12158167001) and Streptavidin conjugated magnetic beads (Life Technologies, Dynabeads M-280 Streptavidin, 112.05D). For co-IPs with MED10B-MEC-ProtA, we used IgG Sepharose 6 Fast Flow (Sigma, GE17-

0969-01) beads and increased washing steps (1× IP buffer, 5× wash buffer, six total). The only modification to buffers was an addition of the detergent NP-40 at 0.1% in the IP and wash buffer. Samples were run on handmade 10% acrylamide SDS-PAGE gels and western blotted accordingly.

## Bimolecular fluorescence complementation

BiFC experiments were performed on 3-week-old *N. benthamiana* plants grown at 22°C under long days (16 hr light/8 hr dark) on soil (Sunshine #4 mix) as per *Martin et al., 2009*. pSITE vectors were used to generate BiFC constructs for MED21, Δ5-MED21, TPL, and TPL$^{H8QuadA}$ – proteins (*Martin et al., 2009*). In all cases, the combinations are N-terminal fusions of either the nEYFP or cEYFP to the cDNA of MED21 or TPL. RFP fused Histone H2B was used as a nuclear marker (*Goodin et al., 2002*). Injection of *Agrobacterium* strains into tobacco leaves was performed as in *Goodin et al., 2002*, but the OD$_{600}$ of the *Agrobacterium* culture used was adjusted to 0.5. Two days after transfection, plant leaves were imaged using an epifluorescence microscope (Leica Biosystems, model: DMI 3000B).

## Chromatin immunoprecipitation and qPCR

Yeast were inoculated in liquid YPD and grown at 30°C with shaking at 225 rpm. After 18 hr of growth, cultures were diluted 1:50 in fresh YPD to a final volume of 200 ml. 4 hr post-dilution, cells were treated with 20 ml of fix solution (11% [vol/vol] formaldehyde, 0.1 M NaCl, 1 mM EDTA, 50 mM HEPES–KOH) for 20 min at room temperature with shaking. Cultures were further treated with 36 ml of 2.5 M glycine for 5 min to quench the cross-linking. Cells were then pelleted at 4°C, washed twice with ice-cold TBS, and flash-frozen in liquid nitrogen. Cells were lysed in breaking buffer (100 mM Tris, pH 8.0, 20% [vol/vol] glycerol, and 1 mM PMSF) on a bead beater before sonication. Samples were processed in a Bioruptor Plus sonication device at 50% for 30 cycles. Following centrifugation (to pellet cellular debris), the supernatant was used for the IP reaction. Biotin-conjugated Anti-HA (High Affinity [3F10], Sigma, 12158167001) coupled to Streptavidin-coated Dynabeads Streptavidin conjugated magnetic beads (Life Technologies, Dynabeads M-280 Streptavidin, 112.05D) was used to probe for HA-tagged TPLN188. Anti-FRB coupled to Protein A Dynabeads (Life Technologies, 100.02D) was used for ChIP on FRB-tagged yeast proteins. Following elution from the beads, samples were incubated overnight at 65°C to reverse cross-links. DNA was purified using a Monarch PCR purification kit (NEB). qPCR for three independent replicates was performed using iQ SYBR Green Supermix (Biorad) in a C100 thermocycler fitted with a CFX96 Real-Time Detection System (Biorad). To calculate fold enrichment, the delta CT (dCT) between input and IP was calculated for each sample for both the control locus (Either ACT1 3′ gene body or a new control primer from a well-characterized gene-free region on Sc chromosome V; *Wong and Struhl, 2011*) and the target locus (i.e., the ARF binding site of the ARC). The delta delta CT (ddCT) was then identified for the (Ct IP) – (Ct control locus) to create the non-specific adjustment. Then the fold enrichment is calculated (2-DDCt).

## Protein alignments

The MED21 protein sequence was aligned to homologs using CLC Sequence Viewer 7, a tree was constructed using a neighbor-joining method, and bootstrap analysis performed with 10,000 replicates.

## Plant growth

For synthetic repression assays in tobacco, *Agrobacterium*-mediated transient transformation of *N. benthamiana* was performed as per *Yang et al., 2000*. 5 ml cultures of *Agrobacterium* strains were grown overnight at 30°C shaking at 220 rpm, pelleted, and incubated in MMA media (10 mM MgCl$_2$, 10 mM MES pH 5.6, 100 µM acetosyringone) for 3 hr at room temperature with rotation. Strain density was normalized to an OD$_{600}$ of 1 for each strain in the final mixture of strains before injection into tobacco leaves. Leaves were removed, and eight different regions were excised using a hole punch, placed into a 96-well microtiter plate with 100 µl of water. Each leaf punch was scanned in a 4 × 4 grid for yellow and red fluorescence using a plate scanner (Tecan Spark, Tecan Trading AG, Switzerland). Fluorescence data was quantified and plotted in R (ggplots). For *Arabidopsis thaliana* experiments using the GAL4-UAS system (*Laplaze et al., 2005*), J0121 was introgressed eight times

into Col-0 accession from the C24 accession and rigorously checked to ensure root growth was comparable to Col-0 before use. UAS-TPL-IAA14$^{mED}$ constructs were introduced to J0121 introgression lines by floral dip method (*Clough and Bent, 1998*). T1 seedlings were selected on 0.5× LS (Caisson Laboratories, Smithfield, UT)+ 25 µg/ml Hygromycin B (company) + 0.8% phytoagar (Plantmedia; Dublin, OH). Plates were stratified for 2 days, exposed to light for 6 hr, and then grown in the dark for 3 days following a modification of the method of *Harrison et al., 2006*. Hygromycin-resistant seedlings were identified by their long hypocotyl, enlarged green leaves, and long root. Transformants were transferred by hand to fresh 0.5× LS plates + 0.8% Bacto agar (Thermo Fisher Scientific) and grown vertically for 14 days at 22°C. Plates were scanned on a flatbed scanner (Epson America, Long Beach, CA) at day 14. *slr* and *med21/MED21* (WiscDsLox461-464K13) seeds were obtained from the Arabidopsis Biological Resource Center (Columbus, OH). CRISPR/CAS9-based mutations in AtMED2 were generated as described in *Wang et al., 2015*. We created a novel mutation in *AtMED21* that introduces a single base-pair insertion of G at nucleotide 214 after the A of the start codon (i214G). This mutation alters the amino acid sequence starting at residue 25 and creates an early stop codon after 11 random amino acids (*Figure 6—figure supplement 1D*).

## Data submissions

All flow cytometry data will be deposited at https://flowrepository.org/. All plasmids will be deposited through Addgene at https://www.addgene.org/Jennifer_Nemhauser/.

## Acknowledgements

We would like to thank Prof. Jef Boeke for kindly providing VEGAS adaptor and regulatory element plasmids; Dr. Jennifer Brophy and Prof. José Dinneny for kindly providing the pUBQ10:GAL4:VP16 plasmid; Dr. Natalia Petrenko and Dr. Kevin Struhl for kindly providing Mediator Anchor Away strains; and Prof. Grant Brown and Prof. Maitreya Dunham for advice on yeast genetics and approaches. We thank members of the Nemhauser group including Amy Lanctot, Romi Ramos, Eric Yang, and Dr. Sarah Guiziou for constructive discussions and comments on this manuscript. We also thank Morgan Hamm for the custom R script used here to analyze Anchor Away plates. Funding: National Institutes of Health (NIH): ARL, HPG, SG, SJS, JEZ, MZ and JLN R01-GM107084. Howard Hughes Medical Institute (HHMI): ARL, SJS, JEZ, MZ, and JLN – Faculty Scholar Award. Ning Zheng is a Howard Hughes Medical Institute Investigator. ARL is supported by the Simons Foundation through the Life Science Research Foundation.

## Additional information

### Funding

| Funder | Grant reference number | Author |
| --- | --- | --- |
| National Institutes of Health | GM107084 | Alexander R Leydon<br>Hardik P Gala<br>Sabrina Gilmour<br>Samuel Juarez-Solis<br>Mollye L Zahler<br>Joseph E Zemke<br>Jennifer L Nemhauser |
| Howard Hughes Medical Institute | | Alexander R Leydon<br>Wei Wang<br>Hardik P Gala<br>Sabrina Gilmour<br>Samuel Juarez-Solis<br>Mollye L Zahler<br>Joseph E Zemke<br>Ning Zheng<br>Jennifer L Nemhauser |

The funders had no role in study design, data collection and interpretation, or the decision to submit the work for publication.

## Author contributions

Alexander R Leydon, Conceptualization, Data curation, Formal analysis, Validation, Investigation, Visualization, Methodology, Writing - original draft, Project administration, Writing - review and editing; Wei Wang, Sabrina Gilmour, Samuel Juarez-Solis, Mollye L Zahler, Joseph E Zemke, Investigation, Visualization, Writing - review and editing; Hardik P Gala, Formal analysis, Investigation, Visualization, Writing - review and editing; Ning Zheng, Resources, Funding acquisition, Writing - review and editing; Jennifer L Nemhauser, Conceptualization, Resources, Supervision, Funding acquisition, Project administration, Writing - review and editing

## Author ORCIDs

Alexander R Leydon (iD) https://orcid.org/0000-0003-3034-1482
Jennifer L Nemhauser (iD) https://orcid.org/0000-0002-8909-735X

## Decision letter and Author response

Decision letter https://doi.org/10.7554/eLife.66739.sa1
Author response https://doi.org/10.7554/eLife.66739.sa2

# Additional files

## Supplementary files

• Source code 1. Custom scripts. Custom scripts used in this study in an R file format to analyze flow cytometry and to quantify root phenotypes. Comments are included to delineate sections in the code. Also available on Github: https://github.com/achillobator/TPL_Structure_Function/.

• Supplementary file 1. Oligonucleotide list. Sequences, names, and experimental uses of all oligonucleotides created in this study.

• Supplementary file 2. Plasmid list. Names and descriptions of all plasmids generated in this study.

• Supplementary file 3. Yeast strain list. Names and full genotypes of all yeast strains generated or used in this study.

• Transparent reporting form

## Data availability

All flow cytometry data is deposited at https://flowrepository.org/. Repository IDS: FR-FCM-Z2GM, FR-FCM-Z2GR, FR-FCM-Z2GQ, FR-FCM-Z2GX, FR-FCM-Z2GT, FR-FCM-Z2W2. All protein interactions have been deposited to IntAct, under the accession code, IM-28972. All code is available through Github: https://github.com/achillobator/TPL_Structure_Function/ (copy archived at https://archive.softwareheritage.org/swh:1:rev:141d7d05fe0c23be55af5050563d160f019d6d65).

The following datasets were generated:

| Author(s) | Year | Dataset title | Dataset URL | Database and Identifier |
|---|---|---|---|---|
| Leydon AR, Nemhauser JL | 2020 | TPL Helix-By-Helix deletion series | https://flowrepository.org/id/FR-FCM-Z2GM | Flow Repository, FR-FCM-Z2GM |
| Leydon AR, Nemhauser JL | 2020 | TPL Helix-By-Helix deletion series 2 | https://flowrepository.org/id/FR-FCM-Z2GR | Flow Repository, FR-FCM-Z2GR |
| Leydon AR, Nemhauser JL | 2020 | TPL Dimerization Mutants 2 | https://flowrepository.org/id/FR-FCM-Z2GQ | Flow Repository, FR-FCM-Z2GQ |
| Leydon AR, Nemhauser JL | 2020 | TPL Dimerization Mutants 1 | https://flowrepository.org/id/FR-FCM-Z2GX | Flow Repository, FR-FCM-Z2GX |
| Leydon AR, Nemhauser JL | 2020 | TPL Helix-By-Helix deletion series 3 | https://flowrepository.org/id/FR-FCM-Z2GT | Flow Repository, FR-FCM-Z2GT |
| Leydon AR, Nemhauser JL | 2020 | TPL Helix-By-Helix deletion series 4 | https://flowrepository.org/id/FR-FCM-Z2W2 | Flow Repository, FR-FCM-Z2W2 |
| Leydon AR | 2021 | Repression by the Arabidopsis TOPLESS corepressor requires | https://www.ebi.ac.uk/intact/ | EBI, IM-28972 |

association with the core Mediator
complex

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
