## [Decision Letter]

**Acceptance summary:**

In this study, Leydon et al., use an elegant multi-component genetic system to address the mechanisms of repression by the Arabadopsis TOPLESS (Tpl) protein. Taking advantage of the genetic tools and knowledge of the structure of the Tpl protein the authors determine two short α helical regions that act as independent repression domains. They provide evidence that the target of one of these domains is the N-terminal region of the Med21 subunit of the mediator complex. Chromatin immunoprecipitation experiments, anchor-away loss of function and co-immunoprecipitation assays indicate that Tpl mediated repression involves formation of a promoter complex comprising the mediator complex along with several general transcription factors, but lacking RNA polymerase II. The authors also show that Tpl-Med21 interactions are involved in Tpl mediated repression in plants.

**Decision letter after peer review:**

[Editors’ note: the authors submitted for reconsideration following the decision after peer review. What follows is the decision letter after the first round of review.]

Thank you for submitting your work entitled "Structure-function analysis of Arabidopsis TOPLESS reveals conservation of repression mechanisms across eukaryotes" for consideration by *eLife*. Your article has been reviewed by 3 peer reviewers, including Irwin Davidson as the Reviewing Editor and Reviewer #1, and the evaluation has been overseen a Senior Editor. The following individual involved in review of your submission has agreed to reveal their identity: Lucia Strader (Reviewer #3).

Our decision has been reached after consultation between the reviewers. Based on these discussions and the individual reviews below, we regret to inform you that your work will not be considered further for publication in *eLife*.

This is an interesting study that uses an elegant multi-component genetic system to address the mechanisms of repression by the Arabadopsis TOPLESS (Tpl) protein. Taking advantage of the genetic tools and knowledge of the structure of the Tpl protein the authors determine two short α helical regions that act as independent repression domains. They provide evidence that the target of one of these domains is the N-terminal region of the Med21 subunit of the mediator complex. They then use the phylogenetic relationships amongst the Tpl proteins to identify a short repression domain in the human TBL1 protein.

Despite defining Tpl repression domains at amino acid resolution and at least in case of Helix 8 one of its targets, the referees were not convinced that the study provided novel insights into the mechanisms of repression. Almost all of the experiments are based on the same AtARC(Sc) system and the cytoSUS interaction assay, but were not supported by more direct biochemical interaction assays or ChIP experiments to address key events in preinitiation complex formation and RNA polymerase II recruitment that may underlie the repressive mechanism. The referees feel that such insight is required for the study to be of general interest to readers of *eLife*.

*Reviewer #1:*

This is an interesting study that uses an elegant multi-component genetic system to address the mechanisms of repression by the Arabadopsis TOPLESS (Tpl) protein. Taking advantage of the genetic tools and knowledge of the structure of the Tpl protein the authors determine two short α helical regions that act as independent repression domains. They provide evidence that the target of one of these domains is the N-terminal region of the Med21 subunit of the mediator complex. They then use the phylogenetic relationships amongst the Tpl proteins to identify a short repression domain in the human TBL1 protein.

Several issues should be addressed.

One of the major conclusions of this study is that the repression domain in helix 8 functions through interaction with the N-terminal domain of Med21. While all of the genetic setup provides strong evidence for this hypothesis, it would have been important to demonstrate this interaction using an independent method, co-immunoprecipitation, recombinant proteins etc. The authors could also address whether Tpl is capable of capturing the entire Med complex or only a sub-complex comprising Med21.

A second major issue is that while the study defines repression domains at amino acid resolution and at least in case of Helix 8 one of its targets, the reader is still left with questions concerning the mechanism of repression. Could the authors perform ChIP studies under the appropriate experimental conditions to assess how the Helix1 and Helix 8 repression domains affect pre-initiation complex formation when they are targeted to a promoter. Do they affect PIC formation Pol II recruitment, promoter release or some other process? It seems the authors have developed a complex and robust genetic reporter system to analyse domains involved in repression. This system can be further exploited to address more mechanistic questions and perhaps even their kinetics. This is particularly important when considering helix 8 and interaction with the Med21. It is not clear to the reader, how this interaction results in repression. This aspect should be further investigated.

*Reviewer #2:*

In this manuscript, the authors studied the specific domains of the plant *A. thaliana* TPL corepressor using a synthetic auxin response circuit (ARC) in the yeast *S. cerevisiae* reported in their previous paper (Pierre-Jerome 2014 PNAS). This artificial system combines plant components including TPL domain fused to IAA domain with auxin-responsive plant promoter and relies on the use of the yeast degradation and transcriptional machinery allowing to monitor the repression and response to auxin of the reporter expression. Based on published structure, the authors identified two domains of TPL corepressor that independently contribute to repression in this system. One of these domains interacts with Med21 Mediator subunit in cytoSUS interaction assay. The authors provide a lot of work with many constructions to study different domains or point amino acid substitutions in TPL-mediated auxin-responsive repression. The question of molecular mechanisms of transcription repression is certainly very important biological question that still remains poorly answered. However, the work did not provide any mechanistic insights and the artificial AtARC(Sc) system and the cytoSUS interaction assay used did not allow to address the repression mechanism in physiological context. Moreover, the role of Mediator complex through its CDK8 kinase module in TPL-mediated repression with a change in Mediator composition previously reported by Furutani laboratory (Ito 2016 PNAS) has not been addressed and integrated with regards to TPL-Med21 interaction.

On my opinion, the work is more suitable for a specialized journal. Many experiments and more direct evidences are needed to support the author conclusions. Without additional evidence, their title that contains "conservation of repression mechanisms" is not supported by experiments. An extensive experimental work should be done to provide at least some mechanistic insights on repression of auxin-responsive gene promoters that could not be achieved within 2 months.

1. The synthetic auxin response circuit (ARC) system is heterological making difficult the interpretation of the results. This artificial system combines plant components including TPL domain fused to IAA domain with auxin-responsive plant promoter and relies on the use of the endogenous, yeast degradation and transcriptional machinery. It was used previously to analyze the implication of different components of auxin response regulation from different plants and their functional annotation or for synthetic biology purposes. However, in this work the use of this system, especially to propose a molecular mechanism of transcription repression and to investigate the implication of Mediator complex is not sufficient and should be completed by additional experiments in plants in more physiological conditions (see below). The authors presented the fact that this system relies on endogenous *S. cerevisiae* proteins of degradation and transcriptional machinery as an argument for conservation of the mechanisms. At the same time, the interaction assay with plant proteins in the yeast is presented as a direct interaction, but one could not exclude the involvement of endogenous proteins in this assay.

2. A direct analysis of gene expression and direct genetic experiments in plants are missing. It should be feasible since many genetic, genomic and biochemical tools are available in *A. thaliana*.

3. The interaction cytoSUS assay is not sufficient to prove the direct interaction between TPL and Med21 Mediator subunit and should be confirmed by additional biochemical experiments with purified proteins or plant crude extracts (for example, by CoIP). In general, TPL interaction with Mediator complex should be analyzed.

4. A direct analysis of Mediator composition, especially with regards to the publication by Ito and colleagues (2016, PNAS) demonstrating the role of Mediator CDK8 kinase module in auxin-responsive repression with TPL-CDK8 module interaction and dissociation of this module for active transcription, is necessary. More generally, the role of this Mediator module in repression mechanisms should be taken into account.

5. The recruitment of transcriptional machinery including Mediator to promoter and regulatory regions of auxin-responsive genes should be analyzed by ChIP. RNA polymerase II recruitment and state should be directly tested in native system.

6. Physiological consequences of the changes in TPL domains and TPL-Med21 interaction should be studied.

7. The results are difficult to follow. First of all, the AtARC(Sc) system should be described at the beginning of the results indicating exactly which construction was used from the previous paper of the authors (Pierre-Jerome 2014 PNAS) and emphasizing the possibility to measure reporter fluorescence to evaluate reporter repression and also the response of the system to auxin (induction in the presence of auxin). The results should be profoundly rewritten for clarity and the figure organization should be readjusted.

For example, why the first figure cited in the results is Figure 1C?

Why some but not all constructions are summarized on Figure 1B with corresponding repression index and auxin induction level? The main figure summarizing all truncations tested would help.

In general, Figure 1 is a complex mix between different panels and constructions.

8. The authors did not provide any direct evidence to claim that "these results indicate that the CRA domain (H3-H8) requires contact with MED21 to drive repression" (p. 12, l. 284). No direct evidence was provided that TPL interaction with Med21 is involved in repression. Med21 is not the only interacting partner of TPL. Moreover, amino acid substitutions in TPL-N188 used in cytoSUS assay that reduced TPL-Med21 interaction also reduced TPL-IAA3 interaction.

9. The protein level of Med21 in wt is lower compared to Med21∆3 and ∆5 (Figure 3B) that should be explained/commented to indicate how this could influence the interpretation of the results.

10. The part of the manuscript on Med21 Anchor Away constructions in AtARC(Sc) system seems very methodological. The combined system is extremely artificial and heterological with three synthetic constructions (AA yeast Med21, inducible yeast Med21, auxin-responsive system with plant components ARF, IAA, TPL-N, plant promoter and endogenous, yeast degradation and transcriptional machinery). There are too many modifications and treatments (rapamycin, auxin, b-estradiol). This is the problem especially since Med21 is a part of the Mediator complex. What happens with Mediator in these conditions? The results are difficult to interpret. The complicated experimental procedure is not clear. The result description is unclear with mistakes (for example, Figure 5E cited on p. 14, l. 371 instead of Figure 4). Why no differences occur between mock and +Rapa for ∆5-MED21-FRB in blue on Figure 4 D? In general, the potential conclusions from this part are not clear.

Alternative strategies, for example point Med21 mutations directly tested in plants should be considered.

11. The part of the results on TPL multimerization and potential conclusions are unclear. It is not well integrated into the manuscript. The authors indicate a caveat in the use of heterological systems that further illustrates the limitations of their use and difficult interpretation of the results. The authors also indicate that "there are important differences between the synthetic and native systems" (p. 19, l. 474). This is the problem for result interpretation. It is not clear why the authors then used another conditional expression system in another plant (tobacco)? (p. 19) Why they use this additional model instead of *A. thaliana* and native gene expression? Finally, they decided to use transgenic Arabidopsis lines and obtained a negative result suggesting that multimer formation is not required for repression.

12. The last part on minimal repression domains in synthetic circuits is not within the general scope of the manuscript. It is not clear why this part was included. One of the possibilities would be to focus the manuscript on synthetic biology purposes and to choose an appropriate journal.

13. Many sentences in discussion are not supported by the results. For example, the authors should not say that their result "suggests a fundamental conservation in at least one corepressor mechanism across species" (p. 22, l. 596). No mechanistic insights were provided and conservation from yeast to plants was not directly addressed. No recruitment experiments were done for TLP, Mediator and RNA polymerase II. No direct experiments were done to demonstrate that TPL-Med21 interaction is directly involved in repression and to propose a molecular mechanism or to integrate their data into previously proposed model for Cdk8 module-containing Mediator. Mediator complex composition and its implication was not directly addressed.

More appropriate references should be cited for the main roles of Mediator complex.

No evidence for stabilization of Mediator complex by TPL were provided.

It is completely incorrect to say that in plants and yeast there is "more primitive form of pausing".

*Reviewer #3:*

In this manuscript, authors seek to resolve conflicting models for corepressor function using the elegant synthetic auxin response system. Auxin signaling is governed by a de-repression paradigm and is ideally suited to interrogate co-repressor function – in this case, the TOPLESS (TPL) co-repressor. Several contradicting models have been put forward for the mechanism of TPL-mediated gene repression, ranging from a requirement for protein oligomerization for activity, interaction with distinct partners, and even which regions of the protein are required for repressive activity. Leydon et al., use the yeast-based synthetic auxin response system to interrogate these models using a single reporter locus, allowing for straight-forward examination of TPL function.

Strengths of this study include the use of a simplified model to study a complex problem and the broad implications for study results (ie, findings likely hold true across organisms). There were a few pieces of data that I found confusing, which I have outlined below. Clarifying these points would strengthen the manuscript.

1. In Figure 1E, it appears that IAA3 alone can repress the reporter to some degree, or at least the maximal reporter activity cannot be achieved with IAA3 present. However, a lack of increased reporter expression with auxin treatment suggests that IAA3 degradation is insufficient to allow for maximal reporter expression (for example, what is seen for the H1-H7 line). I find this to be a curious result; can the authors expand on this?

2. In many cases, experiments to differentiate between variant effects on TPLN activity versus TPLN protein stability are lacking. This data would be helpful for interpreting some of the results. Note: this is included for some assays/variants, but not others.

3. It seems that the images presented for the split ubiquitin assays are stitched together from multiple images. This should be made clear in both the figure legend. In addition, it should be clear whether these were performed as a single experiment or images put together from distinct experiments. If from distinct experiments, I suggest that these be repeated as a single experiment.

4. I don't understand why N188 repression is not relieved by auxin treatment, when both the H1 and H3-H8 truncations are relieved. Maybe this is explained in the text, but if so, I missed this.

5. I also don't understand why H3-H8-QuadAAA is more auxin responsive than H3-H8.

6. I am confused by the data presented in Figure 4D. The text on lines 354-354 suggest that auxin sensitivity of the system is being examined, but it is not apparent that auxin treatment is involved in either the figure or the legend.

[Editors’ note: further revisions were suggested prior to acceptance, as described below.]

Thank you for submitting your article "Repression by the Arabidopsis TOPLESS corepressor requires association with the core Mediator complex" for consideration by *eLife*. Your article has been reviewed by 3 peer reviewers, including Irwin Davidson as the Reviewing Editor and Reviewer #1, and the evaluation has been overseen by a Reviewing Editor and James Manley as the Senior Editor. The following individual involved in review of your submission has agreed to reveal their identity: Lucia Strader (Reviewer #3).

The referees agreed that overall this new version of the manuscript is much improved over the original and many important issues have been addressed. However, a number of serious issues remain. If the authors do decide to submit a revision, it is essential that these concerns are fully addressed.

Essential Revisions:

1. As previously requested, the authors included ChIP experiments to directly show the presence of pre-initiation complex components at the repressed promoter. Nevertheless, it is not clear how the ChIP experiments were quantified, what is enrichment over background, this is not explained in the text. In addition, it is essential to include ChIP controls at an actively transcribed promoter, this is an especially important positive control for RA polymerase II that was not detected at the repressed promoters. Furthermore, Med21 Mediator subunit and TPLN188 were analyzed on chromosome-integrated AtARC locus (Figure 4C) while other ChIP experiments for Mediator subunits, some GTFs and Pol II were done with SPARC plasmid. Chromosomal ACT1 gene body is completely inappropriate as a background for Pol II ChIP, since this region is enriched for Pol II. Appropriate control regions, regulatory, core promoter and transcribed regions, as well as experiments with untagged control strains should be added. Percentage of IP over input values should be presented for untagged control strains and for several regions (negative control, regulatory, core promoter and transcribed regions). Sequences and positions of qPCR primers should be indicated. Mediator is mainly enriched on regulatory (UAS) regions, GTFs are bound on core promoters and Pol II signal in yeast is mostly on transcribed regions. Well-identified UAS enriched by Mediator should be added as positive controls and for comparison. The growth conditions should be clearly indicated to specify that they correspond to repressed state. The ChIP occupancy was analyzed only in transcriptionally repressed state and no results were provided for transition to the active state. This should be analyzed in detail.

2. The co-immunoprecipitation experiments in plant extracts lack a negative control to conclude on the specificity of CoIP signal. For Figure 6B, a control IP without HA tag or antibody should be added to evaluate non-specific binding of YFP-tagged TPL to beads. A condition Med21-YFP-HA "-" YFP-TPL "-" could not serve as a negative control, since no detection with anti-GFP is possible. Using actin depletion as a specificity control is not sufficient and not an accepted control. For CoIP in Figure 6 —figure supplement 1A, a negative control is not appropriate. A control IP is needed to evaluate non-specific binding of YFP-tagged proteins to beads. Furthermore, it would be important to perform co-immunoprecipitation of the Tpl quad AAAA mutant with Med21 to confirm loss of interaction and of Tpl with the N-terminal deletions of Med21.

3. It would be appropriate to remove the dimerization/multimerization experiments either completely or at least relegating them to the supplementary data and move the estradiol inducible experiments (Figure 4 Sup. 3) to main figures. This data appears more pertinent to the message of the paper than the ability of Tpl to form tetramers. Also, the colours in Figure 4 Sup. 3 are difficult to distinguish, can this be improved?

---

## [Author Response]

[Editors’ note: the authors resubmitted a revised version of the paper for consideration. What follows is the authors’ response to the first round of review.]

Reviewer #1:This is an interesting study that uses an elegant multi-component genetic system to address the mechanisms of repression by the Arabadopsis TOPLESS (Tpl) protein. Taking advantage of the genetic tools and knowledge of the structure of the Tpl protein the authors determine two short α helical regions that act as independent repression domains. They provide evidence that the target of one of these domains is the N-terminal region of the Med21 subunit of the mediator complex. They then use the phylogenetic relationships amongst the Tpl proteins to identify a short repression domain in the human TBL1 protein.

We thank the reviewer for their positive comments.

Several issues should be addressed.One of the major conclusions of this study is that the repression domain in helix 8 functions through interaction with the N-terminal domain of Med21. While all of the genetic setup provides strong evidence for this hypothesis, it would have been important to demonstrate this interaction using an independent method, co-immunoprecipitation, recombinant proteins etc. The authors could also address whether Tpl is capable of capturing the entire Med complex or only a sub-complex comprising Med21.

We thank the reviewer for this recommendation, and we have extensively addressed the protein-protein interaction question through the inclusion of several independent methods in both yeast (Co-IP, yeast two hybrid), and plants (BiFC, Co-IP).

A second major issue is that while the study defines repression domains at amino acid resolution and at least in case of Helix 8 one of its targets, the reader is still left with questions concerning the mechanism of repression. Could the authors perform ChIP studies under the appropriate experimental conditions to assess how the Helix1 and Helix 8 repression domains affect pre-initiation complex formation when they are targeted to a promoter. Do they affect PIC formation Pol II recruitment, promoter release or some other process? It seems the authors have developed a complex and robust genetic reporter system to analyse domains involved in repression. This system can be further exploited to address more mechanistic questions and perhaps even their kinetics. This is particularly important when considering helix 8 and interaction with the Med21. It is not clear to the reader, how this interaction results in repression. This aspect should be further investigated.

We fully agree that mechanism of repression is of great interest, and we appreciate the suggestion to investigate the nature of the repressed state. To this end, we performed two types of experiments. First, we performed ChIP-qPCR using Anchor.

Away strains for multiple Mediator subunits, basal transcription factors and RNA Polymerase II--importantly all fused to the same epitope tag. These experiments indicated that: (1) MED21 is indeed associated with the synthetic auxin circuit promoter, and (2) there is significant enrichment of the core Mediator complex and PIC components (minus PolII) at both our synthetic TPL-regulated locus and at a Tup1bound loci (ScSUC1). Second, we used an even more expanded set of strains for Anchor Away analysis in the presence of the Auxin circuit. These experiments connected components from all of the core Mediator subcomplexes to maintenance of transcriptional repression. These two assays complement each other, and greatly strengthen the model that TPL (and likely Tup1) facilitate pre-assembly of the PIC in the absence of PolII, and that this assembly is required for repression.

Reviewer #2:In this manuscript, the authors studied the specific domains of the plant A. thaliana TPL corepressor using a synthetic auxin response circuit (ARC) in the yeast *S. cerevisiae* reported in their previous paper (Pierre-Jerome 2014 PNAS). This artificial system combines plant components including TPL domain fused to IAA domain with auxin-responsive plant promoter and relies on the use of the yeast degradation and transcriptional machinery allowing to monitor the repression and response to auxin of the reporter expression. Based on published structure, the authors identified two domains of TPL corepressor that independently contribute to repression in this system. One of these domains interacts with Med21 Mediator subunit in cytoSUS interaction assay. The authors provide a lot of work with many constructions to study different domains or point amino acid substitutions in TPL-mediated auxin-responsive repression. The question of molecular mechanisms of transcription repression is certainly very important biological question that still remains poorly answered. However, the work did not provide any mechanistic insights and the artificial AtARC(Sc) system and the cytoSUS interaction assay used did not allow to address the repression mechanism in physiological context. Moreover, the role of Mediator complex through its CDK8 kinase module in TPL-mediated repression with a change in Mediator composition previously reported by Furutani laboratory (Ito 2016 PNAS) has not been addressed and integrated with regards to TPL-Med21 interaction.On my opinion, the work is more suitable for a specialized journal. Many experiments and more direct evidences are needed to support the author conclusions. Without additional evidence, their title that contains "conservation of repression mechanisms" is not supported by experiments. An extensive experimental work should be done to provide at least some mechanistic insights on repression of auxin-responsive gene promoters that could not be achieved within 2 months.

As summarized at the top of this document, we took the critiques from Reviewer 2 very seriously, and have added a number of new experiments to address their concerns. Below, we will detail the specific experiments. Please note that we have also changed the title of the manuscript to address the concern that our claims were too bold for our findings.

1. The synthetic auxin response circuit (ARC) system is heterological making difficult the interpretation of the results. This artificial system combines plant components including TPL domain fused to IAA domain with auxin-responsive plant promoter and relies on the use of the endogenous, yeast degradation and transcriptional machinery. It was used previously to analyze the implication of different components of auxin response regulation from different plants and their functional annotation or for synthetic biology purposes. However, in this work the use of this system, especially to propose a molecular mechanism of transcription repression and to investigate the implication of Mediator complex is not sufficient and should be completed by additional experiments in plants in more physiological conditions (see below). The authors presented the fact that this system relies on endogenous *S. cerevisiae* proteins of degradation and transcriptional machinery as an argument for conservation of the mechanisms. At the same time, the interaction assay with plant proteins in the yeast is presented as a direct interaction, but one could not exclude the involvement of endogenous proteins in this assay.

We thank the reviewer for their recommendations. We have extensively addressed the protein-protein interaction question through the inclusion of several independent methods in both yeast (Co-IP, yeast two hybrid), and plants (BiFC, Co-IP). We have also added additional experiments in plants that support the importance of the TPL-MED21 interaction for control of auxin-regulated development.

2. A direct analysis of gene expression and direct genetic experiments in plants are missing. It should be feasible since many genetic, genomic and biochemical tools are available in *A. thaliana*.

We appreciate the reviewer’s desire for more evidence of the relevance of the yeast work for understanding plant biology, and have added a number of experiments with this concern in mind. We have also made many changes to the text to make sure that our claims are stated clearly and conservatively. As both TPL and MED21 play critical roles in a large and diverse set of gene regulatory networks, it is actually quite challenging to address their direct impacts (or the impact of their interaction) on the expression of any one gene or developmental program. To address this significant challenge, we have employed a variety of complementary methods in the synthetic and in the native contexts to accumulate evidence that supports (but could have just as easily have refuted) our model. We have employed more genetic and biochemical assays in the revised text, including cell-type-specific repression and expression lines in plants, as well as BIFC and Co-IP tests for interactions. We did not perform gene expression analysis on these lines, as it would be unlikely to work given that the driver is only expressed in ~5% of root cells (estimate based on Schmidt et al., 2014. The iRoCS Toolbox--3D analysis of the plant root apical meristem at cellular resolution. Plant J 77, 806-14.).

3. The interaction cytoSUS assay is not sufficient to prove the direct interaction between TPL and Med21 Mediator subunit and should be confirmed by additional biochemical experiments with purified proteins or plant crude extracts (for example, by CoIP). In general, TPL interaction with Mediator complex should be analyzed.

As described above, we have extensively addressed the protein-protein interaction question through the inclusion of several independent methods in both yeast (Co-IP, yeast two hybrid), and plants (BiFC, Co-IP). These methods all support the conclusion that TPL and MED21 interact in yeast and in plants. As also described above, we added ChIP and Anchor Away experiments to probe the role of other components of the Mediator complex.

4. A direct analysis of Mediator composition, especially with regards to the publication by Ito and colleagues (2016, PNAS) demonstrating the role of Mediator CDK8 kinase module in auxin-responsive repression with TPL-CDK8 module interaction and dissociation of this module for active transcription, is necessary. More generally, the role of this Mediator module in repression mechanisms should be taken into account.

We agree with the reviewer that the role of the MED13 component of the CDK8 kinase module in TPL-based repression is of great interest. We took two complimentary approaches to probe this relationship. First, we used an Anchor Away experiment in yeast to deplete CDK8 from the nucleus in real time. We found that, in contrast to the marked relief of repression we observed with depletion of core Mediator components, the removal of CDK8 had only a minimal effect. Second, we crossed the dominant effect TPLN188-IAA14 fusion plant lines into the gct-5 background (a MED13 loss of function), and carefully compared the ability of med13 loss of function to influence repression of lateral root development. The choice of med13 allele for these experiments is critical, as other mutations that are in the latter portion of the gene body have a dominant effect in reverting the loss of lateral root phenotype in the solitary root mutant (see Author response image 1). Because the phenotype is dominant (i.e. in gct-2), these alleles are likely reflecting anti-morphic behavior. For this reason, we used the gct-5 allele, which is the most 5’ of all of the available mutations and the most likely to be an amorph.

5. The recruitment of transcriptional machinery including Mediator to promoter and regulatory regions of auxin-responsive genes should be analyzed by ChIP. RNA polymerase II recruitment and state should be directly tested in native system.

We have now included ChIP-qPCR on the yeast circuit to answer this question.

6. Physiological consequences of the changes in TPL domains and TPL-Med21 interaction should be studied.

We agree with the reviewer that understanding the physiological role of the TPL-MED21 interaction is critical (versus other modes of repression through MED13, HDACs, etc). We have included several new experiments that address this concern (Figure 6). The first experiment was to introduce the Helix 8 quadruple point mutant (that abrogates MED21 binding) into the same driver line used for the lateral root suppression assay. The transgenic lines carrying the UAS-TPLN188-Helix8Quad construct produce lateral roots, demonstrating that the Helix 8 residues (and thus TPLMED21 interaction) are required for full repression of target genes. To complement this experiment, we introduced UAS-MED21 with and without N-terminal deletions (Δ3, Δ5, Δ7) into the same driver line. We hypothesized that N-terminal deletions when expressed over the wild-type allele might have a dominant effect on lateral root development. Indeed, we observed an increase in lateral root density compared to wild type. These two reciprocal experiments help to pinpoint the importance of the TPLMED21 interaction in a specific auxin-regulated plant process: lateral root development.

7. The results are difficult to follow. First of all, the AtARC(Sc) system should be described at the beginning of the results indicating exactly which construction was used from the previous paper of the authors (Pierre-Jerome 2014 PNAS) and emphasizing the possibility to measure reporter fluorescence to evaluate reporter repression and also the response of the system to auxin (induction in the presence of auxin). The results should be profoundly rewritten for clarity and the figure organization should be readjusted.For example, why the first figure cited in the results is Figure 1C?Why some but not all constructions are summarized on Figure 1B with corresponding repression index and auxin induction level? The main figure summarizing all truncations tested would help.In general, Figure 1 is a complex mix between different panels and constructions.

We thank the reviewer for this observation, and we have revised the manuscript to focus only on Helix 8 and its interaction with MED21 to allow us to simplify the story. We have also worked to simplify the language and readability of the manuscript overall. Figure 1 is now streamlined accordingly. We have also included more schematics throughout to help guide the reader.

8. The authors did not provide any direct evidence to claim that "these results indicate that the CRA domain (H3-H8) requires contact with MED21 to drive repression" (p. 12, l. 284). No direct evidence was provided that TPL interaction with Med21 is involved in repression. Med21 is not the only interacting partner of TPL. Moreover, amino acid substitutions in TPL-N188 used in cytoSUS assay that reduced TPL-Med21 interaction also reduced TPL-IAA3 interaction.

We have included two experiments that focus on this interaction in planta. First, we performed UAS-TPL-IAA14 experiments with the TPL Helix 8 quad mutations which break TPL’s interaction with MED21. This was sufficient to break the repression of IAA14 controlled genes, and resulted in more lateral root production. Second, we created UAS-MED21 lines with either the wild-type N-terminus, or Nterminal deletions that break TPL-MED21 interaction. These lines show increases in lateral root production, and reciprocally demonstrate that the TPL-MED21 interaction is required for the repression of lateral root development.

9. The protein level of Med21 in wt is lower compared to Med21∆3 and ∆5 (Figure 3B) that should be explained/commented to indicate how this could influence the interpretation of the results.

We understand the concern over the protein level in these assays; however, if anything it is more striking that the protein level of the Med21∆3 and ∆5 are slightly higher and yet still do not interact with TPLN188. This demonstrates that the proteins in this assay are: (a) expressed and stable, and (b) that protein stability is not responsible for the loss of protein interaction.

10. The part of the manuscript on Med21 Anchor Away constructions in AtARC(Sc) system seems very methodological. The combined system is extremely artificial and heterological with three synthetic constructions (AA yeast Med21, inducible yeast Med21, auxin-responsive system with plant components ARF, IAA, TPL-N, plant promoter and endogenous, yeast degradation and transcriptional machinery). There are too many modifications and treatments (rapamycin, auxin, b-estradiol). This is the problem especially since Med21 is a part of the Mediator complex. What happens with Mediator in these conditions? The results are difficult to interpret. The complicated experimental procedure is not clear. The result description is unclear with mistakes (for example, Figure 5E cited on p. 14, l. 371 instead of Figure 4). Why no differences occur between mock and +Rapa for ∆5-MED21-FRB in blue on Figure 4 D? In general, the potential conclusions from this part are not clear.Alternative strategies, for example point Med21 mutations directly tested in plants should be considered.

It was clear that our original submission was challenging to follow and included a number of methods that might not be familiar to all of our readers. We have worked hard to improve readability and accessibility throughout, including adding explanations for why we believed certain approaches were the best choice for answering particular questions. There are a few areas of specific concern here that need clarification. First, the Anchor away system has been extensively utilized in yeast to characterize and understand the mechanism of Mediator’s function at promoters (i.e. PMID:28699889, PMID:24746699). In our case, the only novel component is the Auxin Response Circuit. Therefore, we do not agree that because the circuit is complicated it is not relevant to transcription. We would argue that the dynamic nature of transcription greatly complicates the interpretation of phenotypes of stable point mutations in Mediator components. We strongly believe that our approaches and findings in yeast add unique and highly relevant insights into this complicated process, as well as providing guidance for follow-up work in plants.

11. The part of the results on TPL multimerization and potential conclusions are unclear. It is not well integrated into the manuscript. The authors indicate a caveat in the use of heterological systems that further illustrates the limitations of their use and difficult interpretation of the results. The authors also indicate that "there are important differences between the synthetic and native systems" (p. 19, l. 474). This is the problem for result interpretation. It is not clear why the authors then used another conditional expression system in another plant (tobacco)? (p. 19) Why they use this additional model instead of *A. thaliana* and native gene expression? Finally, they decided to use transgenic Arabidopsis lines and obtained a negative result suggesting that multimer formation is not required for repression.

We have extensively edited the relevant sections of the manuscript to increase clarity of our logic and conclusions.

12. The last part on minimal repression domains in synthetic circuits is not within the general scope of the manuscript. It is not clear why this part was included. One of the possibilities would be to focus the manuscript on synthetic biology purposes and to choose an appropriate journal.

We thank the reviewer for this constructive suggestion, and have removed the discussion of synthetic circuitry in plants from the manuscript

13. Many sentences in discussion are not supported by the results. For example, the authors should not say that their result "suggests a fundamental conservation in at least one corepressor mechanism across species" (p. 22, l. 596). No mechanistic insights were provided and conservation from yeast to plants was not directly addressed. No recruitment experiments were done for TLP, Mediator and RNA polymerase II.

We have now addressed this concern via both ChIP and Anchor away experiments in yeast, as described above.

No direct experiments were done to demonstrate that TPL-Med21 interaction is directly involved in repression and to propose a molecular mechanism or to integrate their data into previously proposed model for Cdk8 module-containing Mediator.

We have now directly tested this in experiments in plants, see previous comment above

Mediator complex composition and its implication was not directly addressed.

We have now included this via both ChIP and Anchor away experiments in yeast, as described above.

More appropriate references should be cited for the main roles of Mediator complex.

We have carefully reviewed the references cited, and would welcome any specific suggestions for further improvement.

No evidence for stabilization of Mediator complex by TPL were provided.

We have addressed this by ChIP which makes it is clear that the Mediator core complex is co-bound with TPLN188 at promoters in yeast.

It is completely incorrect to say that in plants and yeast there is "more primitive form of pausing".This sentence has been removed from the newest version of the manuscript.Reviewer #3:In this manuscript, authors seek to resolve conflicting models for corepressor function using the elegant synthetic auxin response system. Auxin signaling is governed by a de-repression paradigm and is ideally suited to interrogate co-repressor function – in this case, the TOPLESS (TPL) co-repressor. Several contradicting models have been put forward for the mechanism of TPL-mediated gene repression, ranging from a requirement for protein oligomerization for activity, interaction with distinct partners, and even which regions of the protein are required for repressive activity. Leydon et al., use the yeast-based synthetic auxin response system to interrogate these models using a single reporter locus, allowing for straight-forward examination of TPL function.Strengths of this study include the use of a simplified model to study a complex problem and the broad implications for study results (ie, findings likely hold true across organisms).

We thank the reviewer for their positive comments.

There were a few pieces of data that I found confusing, which I have outlined below. Clarifying these points would strengthen the manuscript.1. In Figure 1E, it appears that IAA3 alone can repress the reporter to some degree, or at least the maximal reporter activity cannot be achieved with IAA3 present. However, a lack of increased reporter expression with auxin treatment suggests that IAA3 degradation is insufficient to allow for maximal reporter expression (for example, what is seen for the H1-H7 line). I find this to be a curious result; can the authors expand on this?

We agree that this result gets at a very interesting observation, namely that in circuits where the reporter gene is repressed, it is often activated to a higher level than in control lines where there is no repression (constitutive activation). In our experiments in yeast, IAA3 does not provide any repressive function unless it is fused to TPL, and the increase in activation above this level is due to the switch from repressed to activated state.

2. In many cases, experiments to differentiate between variant effects on TPLN activity versus TPLN protein stability are lacking. This data would be helpful for interpreting some of the results. Note: this is included for some assays/variants, but not others.

This is something that concerned us as well, and we have tried to address it wherever possible. Our hands are tied with respect to the addition of epitope tags in the case of certain experiments. While an epitope tag can be added to the N188 full length protein at the N terminus, the number and length of tags does diminish repression activity. When we use further truncations, the presence of an epitope tag can drastically affect repression. We have tried to include protein levels whenever the inclusion of a tag does not itself change repression strength.

3. It seems that the images presented for the split ubiquitin assays are stitched together from multiple images. This should be made clear in both the figure legend. In addition, it should be clear whether these were performed as a single experiment or images put together from distinct experiments. If from distinct experiments, I suggest that these be repeated as a single experiment.

All experiments within a figure panel are from the same experiment, but were cropped together to conserve space. We have added white lines to indicate this on the panels, and have included text within the figure legend to explain this.

4. I don't understand why N188 repression is not relieved by auxin treatment, when both the H1 and H3-H8 truncations are relieved. Maybe this is explained in the text, but if so, I missed this.

We have included a description of our previous experiments from prior publications that discuss the relevant results that lead to our understanding of why the N188 repressed circuit is not relievable by auxin addition. This is on page 4, lines 78-85.

5. I also don't understand why H3-H8-QuadAAA is more auxin responsive than H3-H8.

We agree that this is an interesting and somewhat surprising result. One way to explain this pattern is that the loss of interaction with Med21 weakens the repression state, as evidenced by higher reporter expression in the absence of auxin. Adding auxin sensitizes the system by decreasing the level of the TPL-fusion, thereby revealing the decrease in stability of this complex and allowing activation at a lower level of auxin. In the context of the entire TPLN188 with the QuadA mutations, we would argue that the second repression domain (H1) and the intact interaction with Med10 are sufficient to maintain a stable repression state, even when the level of TPL is decreased with auxin addition.

6. I am confused by the data presented in Figure 4D. The text on lines 354-354 suggest that auxin sensitivity of the system is being examined, but it is not apparent that auxin treatment is involved in either the figure or the legend.

We thank the reviewer for this correction, which is remedied in the current version of the manuscript.

[Editors’ note: what follows is the authors’ response to the second round of review.]

Essential Revisions:1. As previously requested, the authors included ChIP experiments to directly show the presence of pre-initiation complex components at the repressed promoter.a) Nevertheless, it is not clear how the ChIP experiments were quantified, what is enrichment over background, this is not explained in the text.

We quantified our ChIP experiments using a method applied extensively for such assays in yeast (i.e., in the recent *eLife* paper PMID: 30681409). It is described here: https://www.thermofisher.com/us/en/home/life-science/epigenetics-noncoding-rnaresearch/chromatin-remodeling/chromatin-immunoprecipitation-chip/chip-analysis.html Briefly, the δ CT (dCT) between input and IP is calculated for each sample for both the control locus (Either ACT1 3’ gene body, or a new control primer from a well-characterized gene-free region on Sc chromosome V, PMID: 22156212) and the target locus (i.e., the ARF binding site of the ARC). The δ δ CT (ddCT) is then identified for the (Ct IP) – (Ct control locus) to create the non-specific adjustment. Then the fold enrichment is calculated (2DDCt).

To improve clarity, we have edited the methods section description, and changed the labelling of the y-axis on ChIP graphs in Figure 4 to “Fold enrichment over background (ACT1 Gene body)” or “Fold enrichment over background (Chr.V gene free region)”. We have also included a supplementary ChIP-qPCR figure to assist in guiding the reader (Figure 4 —figure supplement 2).

b) In addition, it is essential to include ChIP controls at an actively transcribed promoter, this is an especially important positive control for RA polymerase II that was not detected at the repressed promoters.

We appreciate this suggestion, and have now tested the enrichment of our proteins of interest at the promoter of the essential plasma membrane ATPase gene *PMA1* (PMID: 3005867). ChIP assays in yeast grown under similar conditions as we are using for our analysis have previously detected enrichment of transcriptional machinery at the *PMA1* locus, and it is frequently used as a housekeeping gene (PMID: 16461706). We tested our ChIP samples at the *PMA1* promoter using validated qPCR primer sets, and detected a significant enrichment of Mediator and GTFs, consistent with previous reports (PMID: 28699889). Thus, we are confident that our ChIP conditions and anti-FRB antibodies are able to capture transcriptionally active loci. The data is shown in Figure 4D. Please see Point D below for experiments examining Pol-II.

c) Furthermore, Med21 Mediator subunit and TPLN188 were analyzed on chromosome-integrated AtARC locus (Figure 4C) while other ChIP experiments for Mediator subunits, some GTFs and Pol II were done with SPARC plasmid.

It is absolutely true that genes on plasmids and those integrated into the genome may behave differently. It is for this exact reason that we carefully validated our results in multiple independent assays (transient and stable transformations, integrated loci and plasmids) in multiple species. We created the SPARC because shared prototrophy genes make it impossible to combine the Anchor Away strains with the integrated ARC as originally constructed. To be scrupulously careful in drawing our main conclusion, we created the integrated strain with TPLN188-HA and MED21-FRB in a non-Anchor Away genetic background to confirm that we saw similar enrichment patterns whether we performed ChIP on MED21-FRB at an integrated locus or on the SPARC plasmid. Because we see that (i) the SPARC behaves similarly to the ARC with respect to repression, and (ii) that we could ChIP MED21-FRB at both the integrated ARC and the plasmid SPARC, we feel comfortable using the SPARC for ChIP assays on other components of the transcriptional machinery.

Given that even this level of caution still left the reviewer with some concerns, we performed additional experimental validations of both the integrated locus and the SPARC plasmid. These results now appear in Figure 4 —figure supplement 2. We find a general similarity in ChIP results between the two reporter locations (SPARC or ARC), and that the ChIP trends track the expression level of the reporter (YFP). Please find the specific description of the experiments in the sections below.

d) Chromosomal ACT1 gene body is completely inappropriate as a background for Pol II ChIP, since this region is enriched for Pol II. Appropriate control regions, regulatory, core promoter and transcribed regions, as well as experiments with untagged control strains should be added.

We have repeated this analysis for the RNA Polymerase II ChIP samples using an alternative primer set. The Chr.V non-genic control primer is the well-characterized gene-free region on Sc chromosome V, utilized in PMID: 22156212. We saw a similar profile for Pol-II at the repressed gene loci at the SPARC reporter promoter and the endogenously Tup1-repressed *SUC1* promoter as compared to the clear enrichment at the promoter of *PMA1*. This indicates that the anti-FRB ChIP was successful, and that the Pol-II is not enriched at the repressed promoters. These data now appear as part of Figure 4D.

e) Percentage of IP over input values should be presented for untagged control strains and for several regions (negative control, regulatory, core promoter and transcribed regions).

We appreciate the reviewer’s suggestion, and have several responses. First, we added an explanation of how ‘Fold enrichment over the background is calculated’ (point A – above, and in the Methods section). We have also shown this graphically by comparing the percent input calculation to the Fold enrichment over background.

This normalization to control allows us to plot ChIPs from different antibodies/IPs onto the same graph. We show this in Figure 4C as it will allow better comparisons to be made with the new Figure 4D, and Figure 4L and the new supplemental ChIP figure (Figure 4 —figure supplement 2).

We have also performed ChIPs with the control strains that do not have any FRB tag present. These data demonstrate that there is no enrichment of the SPARC promoter or relevant control loci (see far left lanes labelled ‘No Tag’ in Figure 4D).

The second part of this question is addressed in our response to point G below, as it is thematically linked.

f) Sequences and positions of qPCR primers should be indicated.

All sequences are included in the provided oligo sequences supplementary file and are labelled with their use as “ChIP-qPCR”. We have updated this list to include all new primer sets. We have also created a map of the ARC promoter, the sequence of which is identical in both the integrated ARC and the SPARC plasmid. This is included in Figure 4 —figure supplement 2.

g) Mediator is mainly enriched on regulatory (UAS) regions, GTFs are bound on core promoters and Pol II signal in yeast is mostly on transcribed regions. Well-identified UAS enriched by Mediator should be added as positive controls and for comparison.

Two pieces of information are important to consider in addressing this issue. The first is that our sonication conditions generate DNA libraries that are centered at ~500 base pairs in size. The second is that the promoter we are analyzing here is very short (363bp, as shown in response to Point G), and indeed this is a similar size to many other genes in the yeast genome. Therefore, the UAS and core promoter cannot be easily untangled without a finer resolution technique such as ChIP-seq, using an alternative fragmentation technique. We tested the resolution of our assays using a series of primers along the gene body, and confirmed that our ‘peaks’ are quite broad (see Figure 2—figure supplement 2). A primer set ~500 base pairs downstream of the ‘UAS’-site still shows a residual enrichment of both TPL and MED21. We will refer to the region we are assaying as the Promoter as opposed to an UAS due to the resolution of our chromatin preparation protocol. Our control gene (the *ACT1* gene body, >700 base pairs downstream of the start codon) is appropriate to test this level of resolution for most cases except for Pol-II, as discussed above. As a note, this figure again highlights why it is critical to normalize the data to a control region. It is only with this step that allows us to plot the data in a single graph, which we think makes for optimal readability and straightforward comparisons.

h) The growth conditions should be clearly indicated to specify that they correspond to repressed state.

We have put headers on the figures to make sure that the growth conditions and repression state are clear. We have also added language to the figure legends to further guide the reader.

i) The ChIP occupancy was analyzed only in transcriptionally repressed state and no results were provided for transition to the active state. This should be analyzed in detail.

We understand the reviewer’s concerns about what the normal enrichment of Mediator / GTFs / Pol-II is at an ARC promoter in the absence of TPL repressor, or in other words what happens at the active ARC promoter. If we understand the reviewer’s comments, we think that one of our statements (“auxin-induced TPL removal changes the composition of Mediator complex”) led to this concern. While we intended to indicate that the concentration of Mediator complex that is actively recruiting Pol-II at the promoter is increasing, we were not clear. We have changed the text to clarify this (Pg.28 lines 656-660). The sentence now reads: “As suggested by Ito and colleagues (Ito et al., 2016) and supported by our synthetic system, auxin-induced removal of TPL is sufficient to induce changes in the activity of the Mediator complex”.

We agree that it is critical to evaluate the change in Mediator abundance at the ARC promoter when it is active. We would first indicate that the original Figure 4L did indeed test this question specifically. We observed a greater enrichment of MED21 at unrepressed promoters. However, to provide additional information, we have added the analysis of the *PMA1* promoter into this figure panel as well as the control, highlighting the lines where there is no repression (No TPL – dark grey). We detect a high abundance of Med21 at the active SPARC promoter, as compared to the repressed promoter. The well-documented *PMA1* promoter has an enrichment of MED21, and is unaffected by the presence of the SPARC. The Δ5Med21 does not fully rescue the association of Med21 to the promoter, however this is roughly similar to the change in the transcriptional output of the SPARC when helix 8 is absent ~1.6x (see transcriptional output in Figure 4J). The Δ5Med21 protein is integrating into Mediator complexes (White bars), however it appears slightly less enriched at the *PMA1* promoter compared to wild-type Med21, supporting the phenotypic analysis that the Δ5Med21 mutant has slower transcription promoting activity (Figure 4 —figure supplement 3A).

To more directly answer the concern over changes in Mediator composition between repressed and active SPARC promoters, we performed ChIP-qPCR in SPARC lines that have no TPL (active). To do this we had to purchase more anti-FRB, which was from a separate lot number from the prior experiments leading to less effective pull-downs. While we observed a lower total enrichment of FRB tagged proteins at all loci compared to the previous experiments (see Figure 4—figure supplement 2D), the trends are entirely consistent. The enrichment of Mediator components MED18 and MED14 at the SPARC promoter is increased under active conditions, whereas CDK8 is absent. Both TBP1 and Pol-II show high levels of enrichment at the active SPARC promoter. The positive control promoter of *PMA1* shows a similar enrichment profile. This indicates that the abundance, and not the composition of the Mediator complex, is altered at the SPARC promoter, with the most striking difference being the recruitment of Pol-II. We have edited our language to make this distinction explicit.

2. The co-immunoprecipitation experiments in plant extracts lack a negative control to conclude on the specificity of CoIP signal. For Figure 6B, a control IP without HA tag or antibody should be added to evaluate non-specific binding of YFP-tagged TPL to beads. A condition Med21-YFP-HA "-" YFP-TPL "-" could not serve as a negative control, since no detection with anti-GFP is possible. Using actin depletion as a specificity control is not sufficient and not an accepted control. For CoIP in Figure 6 —figure supplement 1A, a negative control is not appropriate. A control IP is needed to evaluate non-specific binding of YFP-tagged proteins to beads. Furthermore, it would be important to perform co-immunoprecipitation of the Tpl quad AAAA mutant with Med21 to confirm loss of interaction and of Tpl with the N-terminal deletions of Med21.

We thank the reviewers for their comments on the specificity of the Co-IP signal. We repeated the Co-IPs using the same protocols with the indicated controls. In these experiments, we see no non-specific binding between the TPL-YFP fusion protein and the beads. The Figure 6 panel B has been replaced (New Figure 6B).

3. It would be appropriate to remove the dimerization/multimerization experiments either completely or at least relegating them to the supplementary data and move the estradiol inducible experiments (Figure 4 Sup. 3) to main figures. This data appears more pertinent to the message of the paper than the ability of Tpl to form tetramers. Also, the colours in Figure 4 Sup. 3 are difficult to distinguish, can this be improved?

We thank the reviewers for this suggestion and have given the question of the ‘message of the paper’ a great deal of thought. While the significance of a result is in the eye of the beholder, we see this study as an investigation into the mechanism and functional unit of repression by TPL. While the data on multimerization is not directly relevant to the first of these objectives, it is of central concern for the second. The prevailing thought has been that multimerization is required for repression, and certainly there is data that higher-order multimers-of-multimers exist and have functional roles (PMID: 28630893). To our knowledge, this is the first time that a study has been able to engineer a variant of the TPL N-terminus capable of testing the ability of a monomer of TPL to repress. Therefore, the ability of a monomer of TPL to repress is a striking and novel finding. Other researchers outside of our lab have expressed their interest in this finding as well, indicating to us that these results are integral to the main message of the manuscript. While we appreciate the reviewer’s enthusiasm for the estradiol experiments. We see these results as being supplementary proof that Med21 and TPL interaction has a functional role in repression that is not caused by an off-target effect due to the method of analysis in yeast. Therefore, while we respect the reviewer’s opinion, and have discussed this extensively among the authors, we have left the distribution of main and supplementary figures unchanged.